# MultiFusion: Fusing Pre-Trained Models for Multi-Lingual, Multi-Modal Image Generation

**Marco Bellagente**[4][*][†]   **Manuel Brack**[2,3][†]   **Hannah Teufel**[1][†]   **Felix Friedrich**[3,6]
**Björn Deiseroth**[1,3,6]   **Constantin Eichenberg**[1]   **Andrew Dai**[1]   **Robert J.N. Baldock**[1]
**Souradeep Nanda**[5][*]   **Koen Oostermeijer**[1]   **Andres Felipe Cruz-Salinas**[1]
**Patrick Schramowski**[2,3,6,8]   **Kristian Kersting**[2,3,6,7][‡]   **Samuel Weinbach**[1][‡]

[1]Aleph Alpha, [2]German Research Center for Artificial Intelligence (DFKI),
[3]Computer Science Department, TU Darmstadt, [4]Stability AI, [5]University of Texas,
[6]Hessian.AI, [7]Centre for Cognitive Science, TU Darmstadt, [8]LAION

`marco.bellagente@gmail.com`
`brack@cs.tu-darmstadt.de`
`hannah.teufel@aleph-alpha.de`

## Abstract

The recent popularity of text-to-image diffusion models (DM) can largely be attributed to the intuitive interface they provide to users. The intended generation can be expressed in natural language, with the model producing faithful interpretations of text prompts. However, expressing complex or nuanced ideas in text alone can be difficult. To ease image generation, we propose MULTIFUSION that allows one to express complex and nuanced concepts with arbitrarily interleaved inputs of multiple modalities and languages. MULTIFUSION leverages pre-trained models and aligns them for integration into a cohesive system, thereby avoiding the need for extensive training from scratch. Our experimental results demonstrate the efficient transfer of capabilities from individual modules to the downstream model. Specifically, the fusion of all independent components allows the image generation module to utilize multilingual, interleaved multimodal inputs despite being trained solely on monomodal data in a single language.

## 1   Introduction

The recent popularity of text-to-image diffusion models (DM) [39, 35, 37] can largely be attributed to the intuitive interface they provide to users. The intended generation can easily be expressed in natural language, with the model producing faithful interpretations of a text prompt. Recent works have demonstrated the output quality to be largely dependent on the input encoders with more powerful variants yielding more expressive DMs [39, 2, 12]. We take these insights one step further, vastly enhancing the capabilities of a pre-trained DM through the sophisticated integration of dedicated modules. We propose MULTIFUSION which effectively supports arbitrarily interleaved inputs of multiple modalities and languages. Further, we transfer these capabilities from an underlying language model (LM), eliminating the need for multilingual or multimodal interleaved training data for the generative model. Our approach can utilize readily available datasets and requires less than 5% of the training compute needed to build a comparable DM from scratch.

---

[*]Work performed while at Aleph Alpha
[†]Equal contribution
[‡]Equal supervision

37th Conference on Neural Information Processing Systems (NeurIPS 2023).

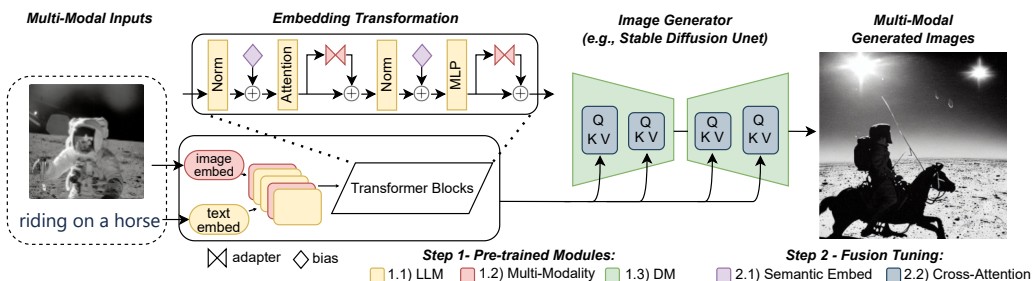

Figure 1: MULTIFUSION architecture. We augment a Decoder Language Model (1.1.) with adapters, finetuned for multimodality (1.2) as well as biases (2.1), finetuned for semantic search. Next, we condition the diffusion model (1.3) through cross-attention (2.2) on embeddings produced by the LM. During diffusion training, we use either the image or the caption for conditioning, while inference is performed with multimodal input. (Best viewed in color.)

The capabilities of current text-to-image DMs are often restricted by the types of inputs they support. MULTIFUSION addresses two of these limitations, facilitating more expressive prompting. Firstly, most of the widely available models are only designed for one language, whereas MULTIFUSION supports five. Contrary to existing multilingual DMs [12], MULTIFUSION requires no multilingual data for DM training and instead only uses readily available English training data. Secondly, text is inherently limited in its expressiveness—many concepts are difficult to articulate with words alone [30]. Consequently, a number of models have been proposed, allowing visual inputs as reference for image generation. Popular examples include image variation [48], adherence to an exemplary style [38], sketch-guided generation [45], or incorporating personal concepts in the resulting images [19]. All of these methods go beyond established image editing techniques [1, 20, 44, 8] where certain aspects of the input picture are altered. However, these *image*-to-image methods are limited to specialized tasks and lack the same level of control and generative versatility for image inputs as for natural language. MULTIFUSION provides a powerful and flexible interface for arbitrary combinations of multimodal and multilingual inputs. The extended prompt capabilities result in a more expressive model facilitating a wide range of complex applications and use cases.

Our main contributions are as follows: (1) We present MULTIFUSION, a multilingual, multimodal diffusion model (2) efficiently bootstrapped using a modular encoder based on an auto-regressive language model. (3) We highlight the use of attention manipulation [14] for multimodal prompting, and (4) demonstrate the transfer of an LM's multilingual capabilities to downstream tasks, removing the need for multilingual downstream training data. (5) Lastly, we introduce the MCC-250 benchmark for the evaluation of compositionality from multimodal inputs.

## 2  Background

**Text-conditioned diffusion.** Diffusion models [39, 35, 2, 37] iteratively denoise a Gaussian distributed variable to produce samples of a learned data distribution. Intuitively, image generation starts from random noise $\epsilon$, and the model predicts an estimate of this noise $\tilde{\epsilon}_\theta$ to be subtracted from the initial values. The denoising process results in a high-fidelity image $x_0$ without any noise. Since this is an extremely hard problem, multiple steps are applied, each subtracting a small amount ($\epsilon_t$) of the predictive noise, approximating $\epsilon$. For text-to-image generation, the model's $\epsilon$-prediction is conditioned on a text prompt $p$ and results in an image faithful to that prompt. The training objective of a diffusion model $\hat{x}_\theta$ can be written as

$$\mathbb{E}_{\mathbf{x},\mathbf{c}_p,\epsilon,t}\left[w_t||\hat{\mathbf{x}}_\theta(\alpha_t\mathbf{x}+\omega_t\epsilon,\mathbf{c}_p)-\mathbf{x}||_2^2\right] \tag{1}$$

where $(\mathbf{x},\mathbf{c}_p)$ is conditioned on text prompt $p$, $t$ is drawn from a uniform distribution $t \sim \mathcal{U}([0,1])$, $\epsilon$ sampled from a Gaussian $\epsilon \sim \mathcal{N}(0,\mathbf{I})$, and $w_t,\omega_t,\alpha_t$ influence image fidelity. Consequently, the DM is trained to denoise $\mathbf{z}_t := \mathbf{x}+\epsilon$ to yield $\mathbf{x}$ with the squared error as loss. At inference, the DM is sampled using the model's prediction of $\mathbf{x}=(\mathbf{z}_t-\bar{\epsilon}_\theta)$, with $\bar{\epsilon}_\theta$ as described below.

Classifier-free guidance [22] is a conditioning method using a purely generational diffusion model, eliminating the need for an additional pre-trained classifier. The approach randomly drops the text conditioning $\mathbf{c}_p$ with a fixed probability during training, resulting in a joint model for unconditional

and conditional objectives. During inference score estimates for the $\mathbf{x}$-prediction are adjusted so that:

$$\tilde{\epsilon}_\theta(\mathbf{z}_t, \mathbf{c}_p) := \epsilon_\theta(\mathbf{z}_t) + s_g(\epsilon_\theta(\mathbf{z}_t, \mathbf{c}_p) - \epsilon_\theta(\mathbf{z}_t)) \tag{2}$$

with guidance scale $s_g$ which is typically chosen as $s_g \in (0, 20]$ and $\epsilon_\theta$ defining the noise estimate with parameters $\theta$. Intuitively, the unconditioned $\epsilon$-prediction $\epsilon_\theta(\mathbf{z}_t)$ is pushed in the direction of the conditioned $\epsilon_\theta(\mathbf{z}_t, \mathbf{c}_p)$ to yield an image faithful to prompt $p$. Lastly, $s_g$ determines the magnitude of the influence of the text $p$. Naturally, the prompt $p$ is not fed directly into the model, but instead a high dimensional representation of $p$ is obtained through a decoder. In prevalent models, the prompt $p$ is text in natural language, whereas in the case of MULTIFUSION, $p$ is a sequence of text and images.

**Multimodality.** Prevalent encoder models like CLIP [34]—or multilingual variants like AltCLIP[12]—are distinctly unsuited for arbitrarily interleaved sequences of images and text. Unlike MULTIFUSION's multimodal encoder, these architectures rely on two separate encoders for textual and visual inputs, with their respective representations being aligned during training. In contrast to our model, the resulting embeddings only encode a single image or textual description but no interleaved sequences comprised of both modalities. Prior work shows that large language models (LMs) produce meaningful representations for conditioning of generative diffusion models [39, 2]. Additionally, pre-trained capabilities of LMs transfer to downstream tasks even without specific finetuning and beyond the initial modalities [9, 29]. SBERT has demonstrated that pre-trained transformer-based LMs can be used to construct encoders for longer contexts [36], albeit exclusively for natural language sequences. Consequently, MULTIFUSION builts its encoder on a pre-trained LM, achieving context-preserving embeddings for multilingual, multimodal inputs.

Other works have used various forms of image conditioning for diffusion models to enable more expressive prompts. Versatile Diffusion [48] enables the generation of image variations through a unified multimodal framework. Rombach et al. [38] proposed retriever-augmented diffusion models facilitating conditioning on particular visual styles provided via exemplary images. Multiple works have introduced methods to generate high-quality, text-conditioned images from low-resolution inputs such as sketches [45, 51]. Furthermore, textual inversion [19] turns concepts from example images into word embeddings that can subsequently be used during generation. This enables incorporating individual styles or objects into generated images. Lastly, Liu et al. [28] proposed a more general approach for diffusion guidance with image inputs using CLIP. Similarly, Bansal et al. [3] applied arbitrary guidance functions to more capable diffusion models. Such methods facilitate image generation, for example, from segmentation maps, image variations, or style transfer. However, this type of guidance requires a separate model and more complex, hand-crafted pipelines. In contrast, MULTIFUSION introduces a unified, general pipeline for effective direct conditioning through classifier-free guidance, removing the need for separate components. Concurrent with our work, GlueGen also attempt the task of aligning additional input modalities to pre-trained text-to-image models [33]. We encourage the reader to refer to their work for more details.

**Multilingualism.** Existing LMs pre-trained on multilingual data show impressive multilingual capabilities [40, 26, 13, 49]. Popular text-to-image DM's, however, are usually designed for a single input language [39, 37, 2, 17]. AltDiffusion [12] addressed this issue by proposing a multilingual text encoder (AltCLIP) on which the DM is conditioned instead. AltCLIP embeddings are aligned to the previously used CLIP encoder in a contrastive teacher-student setup using a large multilingual corpus. Subsequently, the cross-attention layers of a pre-trained Stable Diffusion model are finetuned to utilize the AltCLIP encoder instead. The resulting AltDiffusion model can be prompted in 9 different languages. AltDiffusion's image generation is aligned so that the same prompt in different languages results in similar images. With MULTIFUSION we leverage cross-lingual transfer [13, 49] to enable multilingual prompting. More precisely, our generative model obtains multilingual capabilities from the encoder despite being solely trained on English data.

## 3 MULTIFUSION

By fusing pre-trained model components, MULTIFUSION creates one cohesive system that requires less than 5% of the computational resources needed to train a comparable model from scratch. Our approach involves replacing the encoder of Stable Diffusion (SD) with a more advanced one built on a pre-trained LM. Fusing these components results in a downstream model with the ability to comprehend multilingual, interleaved multimodal inputs. The image generation component inherits this potential despite being trained solely on mono-modal data in a single language. The architecture

of MULTIFUSION is illustrated in Fig. 1. The vast majority of pre-trained weights remain frozen, resulting in an efficient computational process overall. Further information on the training data sizes, parameter counts, and GPU hours for all components is supplied in Tab. 3, along with details on data splits, languages, and modalities in Tab. 4 of the appendix. Subsequently, we outline how to combine and align the involved modules for image generation effectively.

**Input encoders.** The CLIP encoder [34] used by SD is unsuited for interleaved multimodal inputs as it disregards context and yields disjoint encodings of text and images. Previous work has demonstrated that text encoders based on context-sensitive LMs improve the expressiveness of downstream image generation models [39, 2]. Accordingly, we model the backbone of MULTIFUSION's encoder as an autoregressive transformer [10] using rotary position embeddings [43] trained on a multilingual corpus of various languages (step 1.1 in Fig. 1). We chose an autoregressive decoder model over a bi-directional architecture since decoder models intuitively outperform bi-directional models on relevant tasks. For example, autoregressive models excel at manipulation with natural language ("Subject X with background Y") (cf. Sec. 4.2) or correct features attributions ("Red X, Blue Y") (cf. Sec. 4). Previous research has identified the natural breaking of permutation equivariance as the source of these capabilities [24], compared to bidirectional models relying entirely on positional embeddings. We acknowledge that bi-directional models may outperform autoregressive ones on other embedding tasks [50], but argue that an autoregressive model is better suited for the tasks studied in MULTIFUSION due to the outlined benefits.

Following the methodology proposed by MAGMA [15], we consider an LM with an added image prefix and dedicated adapters to enable multimodal capabilities (step 1.2 in Fig. 1). Adapters are a suitable architectural choice for multimodal prompts since previous research has already performed extensive ablations on adapter architectures and demonstrated their improved understanding of multimodal inputs over other methods [15]. In this architecture, the image prefix maps the image into sequences of token embeddings in a joint multimodal input embedding space. The adapters are added to each attention and feed forward layer of the transformer and are trained autoregressively on a combination of large-scale image-text datasets (cf. App. A), while the parameters of the language model remain frozen [15, 29, 23]. As a result, the LM enables prompting with arbitrarily interleaved sequences of text and image tokens.

**Semantic embeddings.** In order to use the LM as an encoder for image generation, we extract the model's hidden representations before the language modeling head. While these representations already capture semantic information from the model's pre-training, they need to be optimized and aligned further for usage in arbitrary downstream tasks (step 2.1 in Fig. 1). Initial experiments without additional alignment have shown low rates of convergence (based on visual inspection of generated outputs). Consequently, we chose to produce semantic embeddings guided by the intuition that a focus on the semantics of a text prompt would best capture the information relevant to image generation. Thus simplifying the learning of mapping from embeddings to image outputs, which was confirmed by our initial experimental observations. In conclusion, we deem semantic fine-tuning an essential condition for successfully fusing an image generation model.

We obtained high-quality semantic embeddings through parameter-efficient bias fine-tuning [4]. The tuning follows the supervised contrastive learning objective outlined in S-GPT [31]. The training data consists of natural language tuples of premise and hypothesis, where entailments serve as positive samples and contradictions as negative ones. Importantly, the tuples are bi-lingual such that the premise and hypothesis are each stated in different languages. We observed that including only two languages in this finetuning task is sufficient to achieve multilingual alignment even beyond bilingualism (cf. Sec. 4). Crucially, the semantic bias weights are tuned independently of the multimodal adapters, allowing for modular extensions of the LM with both components.

**Bootstrapping Stable Diffusion.** The final MULTIFUSION encoder is the result of combining these separately trained modules (cf. Fig. 1). The DM is conditioned on embeddings extracted from the last hidden layer of the transformer. Subsequently, we denote $H(x)$ as embedding for input $x$ after a forward pass through MULTIFUSION's multimodal LM encoder. We now align the pre-trained image generation model with MULTIFUSION's encoder (step 2.2 in Fig. 1). Considering the depicted SD architecture, we only need to alter the conditioning portion of the generative model to use our new encoder instead of CLIP. In line with previous research [12], we keep all weights of the DM frozen and only finetune the cross-attention layers of the U-Net. The training objective remains the same as shown in Eq. 1 and 2 with the conditioning $\mathbf{c}_p$ being the encoding $H(x)$.

Table 1: FID-30k and ClipScores on the MS-COCO validation set for MULTIFUSION (MF) and Stable Diffusion (SD). SD is always prompted with text. All images were generated with image size 256x256 and 100 diffusion steps. Textual prompts consisted of the COCO image caption, multimodal prompts of the caption and COCO reference image, and image prompts of just the image.

| | **FID-30K** $\downarrow$ | | | | **CLIPScore (Text-to-Image)** $\uparrow$ | |
|---|---|---|---|---|---|---|
| Guidance Scale | SD v1.5 | MF (Text) | MF (Multimodal) | MF (Image) | SD v1.5 | MF |
| 8.0 | 14.62 | 14.19 | 11.50 | 8.02 | 0.31 | 0.30 |
| 6.0 | 12.73 | 12.15 | 10.29 | 7.18 | 0.31 | 0.29 |
| 4.0 | 10.19 | 9.90 | 8.53 | 6.03 | 0.31 | 0.29 |
| 2.0 | 9.74 | 12.21 | 8.61 | 6.05 | 0.30 | 0.28 |
| 1.0 | 26.09 | 32.81 | 24.22 | 18.93 | 0.27 | 0.25 |

We trained the DM only on monolingual (English) and monomodal inputs with $x$ being randomly chosen as either an image *or* a caption. Nonetheless, the final MULTIFUSION model is capable of interpreting multilingual and arbitrarily interleaved text and image prompts. Our method highlights that the capabilities of strong LMs can be transferred to downstream tasks without the need for dedicated downstream training data. We provide empirical evidence in the following section.

**Modality alignment.** During initial experiments, we observed that further modifications to the inference mechanics are needed to enable stable multimodal prompting. More specifically, for interleaved text and image prompts MULTIFUSION's encoder represents one image as a sequence of 144 tokens in the input embedding space. In most scenarios, the accompanying text prompt contains fewer tokens, resulting in a disproportional influence of the visual inputs on the generated image. To counteract this phenomenon, we utilize attention manipulation [14] to up-weight the impact of textual tokens with respect to the discrepancy in input length. Representative results showcasing the effect of attention manipulation can be found in Appendix D.

## 4   Experiments

Next, we present exhaustive evaluations of the multimodal and multilingual capabilities of MULTIFUSION on various benchmarks. To evaluate compositional robustness on models allowing multi-modal inputs, we introduce the new MCC-250 benchmark. Furthermore, we showcase a variety of applications enabled by MULTIFUSION. We provide a detailed experimental protocol, including information on implementation and training setup in App. A.

### 4.1   Empirical evaluation

**Image fidelity & image-text alignment.** We start off with a standard evaluation for image generation models to demonstrate the general functionality of MULTIFUSION. We investigate image fidelity using FID-30k scores on the MS COCO validation set [25]. We report the results using textual, multimodal, and image prompts and a comparison against SD v1.5 in Tab. 1. The results show that the image fidelity of MULTIFUSION with textual prompts is competitive with the underlying SD model. Improved FID scores highlight that the capabilities of the original model are preserved in addition to benefiting from new input modalities and languages.

Using multimodal inputs instead of textual prompts results in a substantial improvement of FID scores. This improvement clearly indicates that visual inputs possess a greater capacity to encode comprehensive information about the underlying distributions, surpassing the effectiveness of textual descriptions alone. We acknowledge that using the MS COCO reference images as prompts provides a strong supervision. However, we argue that the above conclusion of image inputs adding additional and more fine-grained information over text prompts alone still holds. Because MULTIFUSION achieves the improved FID scores by generating meaningful variations of the prompt with more aligned details instead of just trivially reproducing the input image (cf. Sec. 4.2 and App. B). Beyond FID, we evaluated text-image alignment as average CLIPScore [21] between COCO captions and generated images. Again, MULTIFUSION achieves scores on par with SD, confirming the preservation of previous capabilities.

Table 2: Fine-grained human evaluation results on MCC-250. SD and Composable Diffusion were prompted with text, whereas we show results for both text and multimodal prompts for MULTIFUSION. Specifically, multimodal prompts contain one visual reference for each object interleaved with the text input. Recall that each prompt is a complex conjunction of two different objects with different colors. MULTIFUSION with multimodal inputs strongly outperforms both baseline models.

| Methods | Zero obj ↓ | One obj. ↑ | One obj. w/ correct color ↑ | Two obj. ↑ | Two obj. w/ correct colors ↑ |
|---|---|---|---|---|---|
| Stable Diffusion [%] | $0.92_{\pm 4.81}$ | $99.07_{\pm 4.89}$ | $90.01_{\pm 13.97}$ | $44.89_{\pm 28.61}$ | $29.92_{\pm 24.76}$ |
| Composable Diffusion [%] | $3.88_{\pm 7.49}$ | $96.01_{\pm 7.72}$ | $88.49_{\pm 42.83}$ | $34.72_{\pm 22.79}$ | $25.59_{\pm 18.94}$ |
| MultiFusion (text) [%] | $1.08_{\pm 4.55}$ | $98.91_{\pm 4.57}$ | $82.36_{\pm 18.98}$ | $36.03_{\pm 29.17}$ | $21.66_{\pm 22.23}$ |
| MultiFusion (multimodal) [%] | $\mathbf{0.55}_{\pm 2.81}$ | $\mathbf{99.44}_{\pm 2.85}$ | $\mathbf{94.88}_{\pm 11.37}$ | $\mathbf{65.06}_{\pm 30.64}$ | $\mathbf{58.35}_{\pm 30.94}$ |

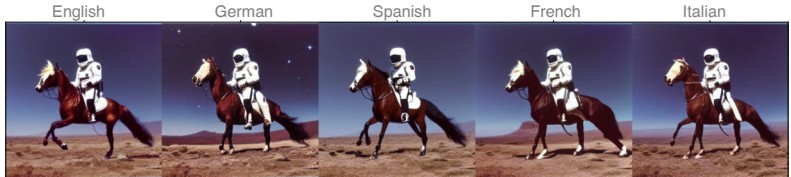

Figure 3: Multilingual alignment of images generated by MULTIFUSION. All images were generated using the same seed and with the respective translation of the prompt '*an image of an astronaut riding a horse*'. (Best viewed in color)

**Compositional robustness.** A challenging task many image synthesis models struggle with is image composition. DMs and, in particular, SD often fail to correctly compose the objects and attributes specified in the text prompt [11, 16].

MULTIFUSION, on the other hand, behaves robustly with respect to challenges in compositional generation as shown in Fig. 2. For evaluation, we propose a new benchmark, Multimodal Concept Conjunction 250 (MCC-250)[4]. It builds on a subset of the Concept Conjunction 500 (CC-500) benchmark [16]. CC-500 contains 500 text prompts of the pattern "a red apple and a yellow banana", textually describing two objects with respective attributes. For half of those prompts, we curated a set of images for each object, enabling multimodal prompting, i.e. the textual description is interleaved with a visual reference. For the new MCC-250 benchmark, we present a human evaluation of SD, Composable Diffusion [27] and MULTIFUSION in Tab. 2. Note that all approaches use the same image generation module. For MULTIFUSION we evaluate multimodal prompts (text and image) as well as text-only.

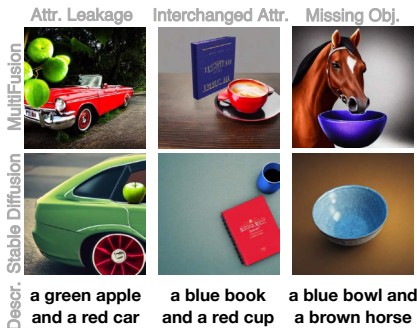

Figure 2: The multimodality of MULTIFUSION proves more robust for image compositions. SD is prompted in text with '*A photorealistic image of {Descr.}*'. MULTIFUSION prompts contain interleaved visual references. (Best viewed in color.)

On this complex task, MULTIFUSION clearly outperforms both baseline models, almost doubling the portion of images containing both objects with correct colors. Successful compositions once again emphasize the capacity of multimodal prompting in MULTIFUSION. Importantly, the observed performance improvement originates from the multimodality of inputs, as the success rate on text-only inputs remains comparable. Surprisingly, Composable Diffusion performs slightly worse than SD for image composition. The resulting images frequently display strange blends of the two objects rather than capturing them as distinct entities. We provide further details on the user study in App C.

**Multilingual alignment.** Next, we investigate the multilingual capabilities of MULTIFUSION. Therefore, we evaluated the alignment of directly translated text inputs in prompt embedding space and the generated images. We compare MULTIFUSION with AltDiffusion [12], as it also builds on a frozen SD model. AltDiffusion makes for a great comparison, as the key differences between both

---

[4]We make the MCC-250 benchmark available at `https://huggingface.co/datasets/AIML-TUDA/MCC-250`

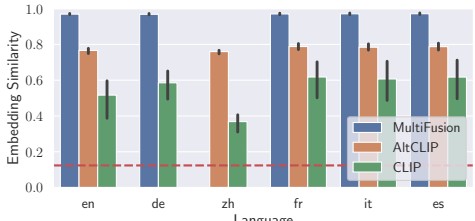 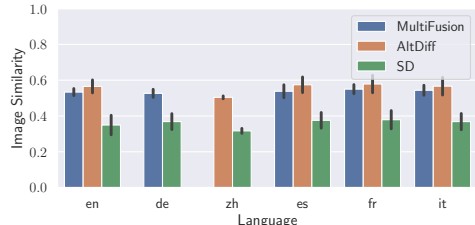

(a) Textual embedding alignment as average cosine similarity between prompt translations. Error bars indicate the standard deviation across paired language comparisons. We provide scores for CLIP as a reference where the horizontal red line indicates the lower bound of average CLIP similarities between uncorrelated english prompts.

(b) Alignment of generated images using average normalized multi-scale structural similarity scores [46] over 10 generated images per prompt and language. Error bars indicate the standard deviation across languages. SD is included for reference.

Figure 4: Comparison of multilingual alignment over DrawBench prompts. MULTIFUSION achieves comparable alignment of the output images although the image generation module was only trained on English data. This can be attributed to the strong alignment of multilingual prompts in MULTIFUSION's embedding space. Similarities are calculated based on paired comparisons between one language and all others. We do not report German results for AltClip/AltDiff nor Chinese for MultiFusion as these languages are not in the respective training data.

models lie in the respective encoders and training data. AltDiffusion uses a large multilingual dataset to finetune the DM, whereas MULTIFUSION's finetuning requires only English training data.

The evaluation is based on a multilingual version of DrawBench [39][5]. To that end, we translated all prompts into the secondary languages of MULTIFUSION and AltDiffusion: German and Chinese. Additionally, we include the three languages shared by both models: French, Spanish, and Italian. In Fig. 4a, we plot the alignment of multilingual text embeddings over DrawBench. MULTIFUSION's encoder clearly outperforms AltDiffusion's encoder (AltClip) on embedding alignment, scoring 97% cosine similarity on average. The alignment of the generated images is similar for both models, although MULTIFUSION was only finetuned using English data (cf. Fig. 4b). These results highlight that good multilingual embedding alignment enables the transfer of multilingualism to downstream tasks without the need for explicitly multilingual training data.

## 4.2 Applications

After we empirically demonstrated MULTIFUSION's multimodal, multilingual prompting capabilities, we show how it facilitates diverse interesting use cases. We provide more examples in the Appendix.

**Image composition.** One of the main strengths of MULTIFUSION's multimodal inputs is image composition. As demonstrated in our empirical evaluation and throughout Figs. 5a and 5b, text and image sequences can be combined arbitrarily and flexibly. Interleaved inputs allow for intuitive combinations of images (cf. Fig. 5b) and serve as a visual reference to enrich text prompts.

**Negative prompting.** Negative prompting refers to a technique in image generation that aims to suppress certain concepts. To that end, the unconditioned estimate during classifier-free guidance $\epsilon(z_t)$ (cf. Eq. 2) is replaced with an estimate conditioned on a negative prompt $\epsilon(z_t, c_n)$. Guiding away from $\epsilon(z_t, c_n)$ results in the concepts described by $n$ being avoided. Consequently, negative prompting works better with more descriptive prompts $n$. Our previous results demonstrated (cf. Tab. 1) a higher expressiveness of images over text that also translates to negative prompts. Fig 5c shows that textual negative prompts are less effective in removing undesired concepts. However, using image prompts, these concepts can be completely suppressed.

**Image variation.** MULTIFUSION offers a direct interface for generating image variants. Simply providing a single image as input prompt to the default pipeline produces meaningful variations already. Other models, in contrast, rely on inversion or re-noising of the input image. We depict

---

[5]We did not include prompts from the categories 'misspellings' and 'rare words'.

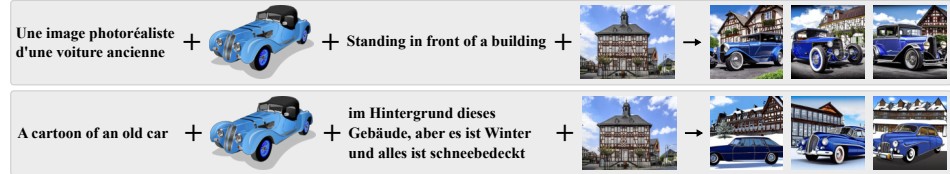

(a) Multilingual, multimodal prompting. MULTIFUSION prompts seamlessly integrate arbitrary combinations of images and textual prompts in multiple languages.

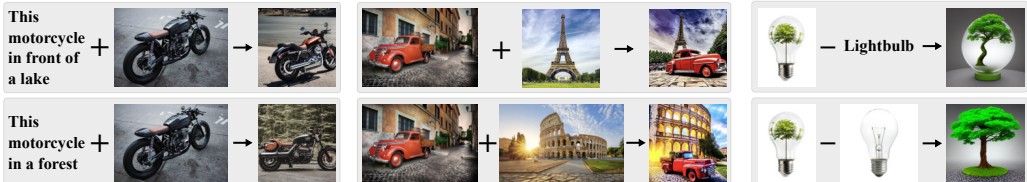

(b) Multimodal Image Composition.          (c) Negative Prompting.

Figure 5: Applications of MULTIFUSION highlighting the versatility and expressiveness of multilingual, multimodal prompts. (Best viewed in color)

examples in Fig. 6a. The generated images include sensible deviations from the input but faithfully reproduce the underlying image contents.

**Style modification.** Furthermore, MULTIFUSION can easily generate images adhering to any artistic style. The style of images is notoriously hard to describe in image generation prompts and has even led to the development of monetized prompt databases for long, obfuscated prompts. This is mostly due to the magnitude of factors that have to be accounted for, such as the color palette, composition, contrast, etc. Fortunately, all of these aspects can easily be expressed through exemplary (reference) images. Fig. 6b depicts examples where arbitrary styles are applied to various scene descriptions. MULTIFUSION delivers high-quality outputs that show the prompted scene in the desired style.

**Multilingualism.** Lastly, the strong multilingual embedding alignment makes MULTIFUSION largely invariant to the language of input prompts. Our model fully supports input prompts in five languages: English, German, French, Spanish, and Italian. In line with our results on language alignment, Fig. 3 shows that the same prompt in different languages yields largely similar images. This improves accessibility and expands the group of users, regardless of their native language. Moreover, Fig. 5a demonstrates that languages can even differ within the same input. This further emphasizes MULTIFUSION's expressiveness, allowing users to construct prompts in various ways.

## 5    Discussion

The capabilities of MULTIFUSION outlined above emphasize the advantages of the model, which we subsequently discuss alongside the remaining limitations and overall societal impact.

### 5.1    Advantages of MULTIFUSION

Multimodality and multilingualism of inputs offer various benefits compared to prevalent models. The extension of the interface beyond the usual English-only text-to-image applications makes MULTIFUSION more flexible, expressive, and versatile.

**Expressiveness.** Natural language and images both have benefits and limitations in the concepts they can convey. When used in combination, however, we are able to draw from the advantages of one and the other, thus eliminating the restrictions of either modality. On the one hand, complex objects that should be included can easily be prompted via exemplary images instead of using long, convoluted, and error-prone textual descriptions. Natural language is often unclear and ambiguous and may lack words to describe certain concepts concisely. For the car depicted in Fig. 5a, for example, MULTIFUSION faithfully infers the vehicle's color, make, type, and era and uses that as a reference for image generation. Achieving similar results through textual descriptions alone requires complicated prompt engineering and may be unsuccessful altogether. Additionally, some necessary

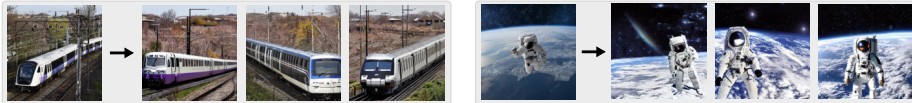

(a) Image Variations. MULTIFUSION produces non-trivial variations when prompted with a single image.

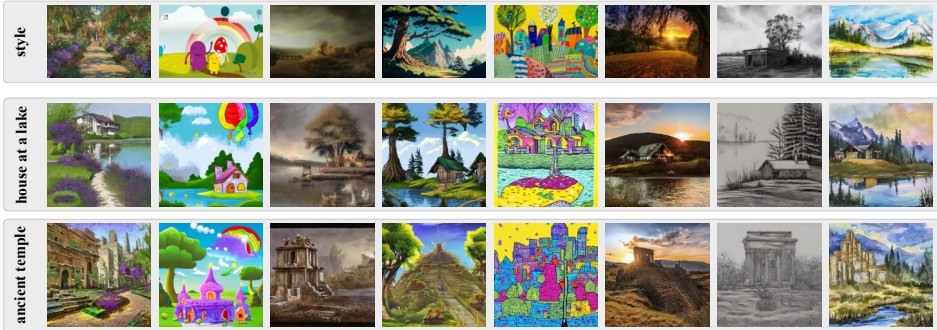

(b) Style transfer from reference images. MULTIFUSION is prompted with a text prompt (left) + '*in the style of <image>*'. Generated images are faithful to the targeted scenario and style.

Figure 6: Further applications of MULTIFUSION. (Best viewed in color)

terminology may elude the end user. For example, a non-native speaker may not be aware that the type of building in Fig. 5a can be described as *half-timbered*. Users may consequently write sub-optimal textual descriptions which can be circumvented using visual prompts. Furthermore, images convey world knowledge that the model does not possess. For example, not all existing artworks or paintings are included in the training data and retained by the model (cf. Fig. 6b). Consequently, using their stylistic characteristics as reference is difficult through text alone. MULTIFUSION easily bridges this knowledge gap by facilitating the inclusion of these images in the prompt. This even enables using individual experiences for reference, such as my car, a sketch I drew, or a building I saw. On the other hand, natural language provides an intuitive interface for abstract concepts and scenarios. For example, the drawing of the car in Fig. 5a is easily turned into a photorealistic version through textual instruction. Consequently, the capabilities of MULTIFUSION result from combining the expressiveness of images and the abstraction of natural language. The inclusion of multiple languages only further increases the model's expressiveness.

**Robustness and clarity.** Similarly, the availability of multimodal inputs significantly improves the robustness of MULTIFUSION. The combination of visual and natural language inputs allows users to circumvent the limitations of the modalities themselves and those of current DMs. For example, natural languages contain a variety of homographs, i.e., words with the same spelling but different meanings. Inferring the correct one from context is often impossible, whereas images provide clear examples. On the other hand, images alone may contain various sources of noise, such as background objects or stylistic choices that the user might not want to consider for generation. However, when used in combination, one modality is able to make up for the shortcomings of the other.

Furthermore, MULTIFUSION overcomes the limitations of current DMs in compositional generation [11, 16]. As demonstrated on the MCC-250 benchmark, images generated from multimodal prompts with MULTIFUSION are less likely to miss objects specified in the prompt or bind attributes incorrectly. This is due to MULTIFUSION devoting multiple tokens to an object described via an image and using context-aware embeddings. In contrast, text-only generation using models such as Stable Diffusion often leads to regular attribute leakage or interchanged attributes, cf. Fig. 2. In this case, the green color of the apple leaks to the car, or colors are mixed up between objects. We include further examples in App. C showcasing the resilience of MULTIFUSION.

**Licensing.** Recently, copyright issues and licensing of training images have been heavily discussed topics[6]. Current legal actions[7] may lead to a variety of images—especially creations from artists—being no longer admissible for use in training. Consequently, applications such as instructing the

---

[6] https://www.copyright.gov/rulings-filings/review-board/docs/a-recent-entrance-to-paradise.pdf
https://www.govinfo.gov/content/pkg/FR-2023-03-16/pdf/2023-05321.pdf
[7] https://stablediffusionlitigation.com

model to reference an artist's style will be infeasible even though the end user might have the necessary copyright permissions. With MULTIFUSION, however, a user can still provide reference material not included in the training data simply in the input.

## 5.2 Limitations

While MULTIFUSION achieves impressive results on various applications, there are some limits to the model's expressiveness. For one, MULTIFUSION always produces meaningful variations when prompted with a single input image. While this is an advantage in some settings, it may be a limitation in others since exactly copying objects from an input is not feasible. This originates from the encoder not being designed to reconstruct images from its representations but to encode relevant characteristics efficiently. Nonetheless, this behavior is generally intended as images are supposed to serve as references to MULTIFUSION instead of performing image editing. Additionally, the quality and composition of an input image have a significant effect on the generated image. We observed that some visual inputs must be carefully selected to achieve the desired outcome. For example, MULTIFUSION may also include unwanted items from the background of an image in the generated output. Furthermore, in some cases, MULTIFUSION has a tendency to copy the input image style even when the prompt's textual portion indicates a different one. Nonetheless, this can be adequately addressed with the proposed attention manipulation method [14], as demonstrated throughout the examples in this work.

## 5.3 Societal impact

Recent developments in text-to-image models [35, 32, 39, 2] have the potential for a far-reaching impact on society, both positive and negative when deployed in applications such as image generation, image editing, or search engines. Previous research [41, 18] described many potential negative societal implications that may arise due to the careless use of such large-scale generative models. Many of these problems can be attributed to the noisy, large-scale datasets these models rely on. Since recent text-to-image models, such as SD, are trained on web-crawled data containing inappropriate content [42, 6, 5], they are no exception to this issue. Specifically, models relying on the LAION datasets [42] show signs of inappropriate degeneration [41]. Consequently, we assume MULTIFUSION to suffer from similar shortcomings only reinforced by its additional capabilities. Therefore, we will not make the model weights publicly available in their current form.

# 6  Conclusion

In this work, we introduced MULTIFUSION, a diffusion model utilizing multilingual, arbitrarily interleaved multimodal inputs. These capabilities provide a versatile interface to users in order to better express themselves. We demonstrated that multilingual alignment in the encoder is sufficient to achieve multilingualism in downstream tasks, eliminating the need for dedicated multilingual datasets. Thus, significantly reducing computational requirements. More generally, MULTIFUSION highlights how separate pre-trained components can be interconnected to realize complex models.

Numerous promising avenues for future research emerge from our current work. One intriguing direction involves expanding the scope of MULTIFUSION to facilitate interactive image generation through integration with a chat interface. This novel extension would offer an enhanced user experience, introducing a dynamic interplay between the system and users. Additionally, we envision the potential for recursive prompting of generated images, enabling a progressive refinement of output through incremental instructions. Moreover, the incorporation of an additional decoder, based on the MULTIFUSION encoder, holds promise for facilitating textual generation resulting in multimodal outputs. Lastly, we propose extending the encoder itself to encompass an even broader range of modalities, including audio, video, time series, and various others.

## Acknowledgments

We gratefully acknowledge support by the German Center for Artificial Intelligence (DFKI) project "SAINT", the Federal Ministry of Education and Research (BMBF) project "AISC " (GA No. 01IS22091), and the Hessian Ministry for Digital Strategy and Development (HMinD) project "AI

Innovationlab" (GA No. S-DIW04/0013/003). This work also benefited from the ICT-48 Network of AI Research Excellence Center "TAILOR" (EU Horizon 2020, GA No 952215), the Hessian Ministry of Higher Education, and the Research and the Arts (HMWK) cluster projects "The Adaptive Mind" and "The Third Wave of AI", and the HMWK and BMBF ATHENE project "AVSV".

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

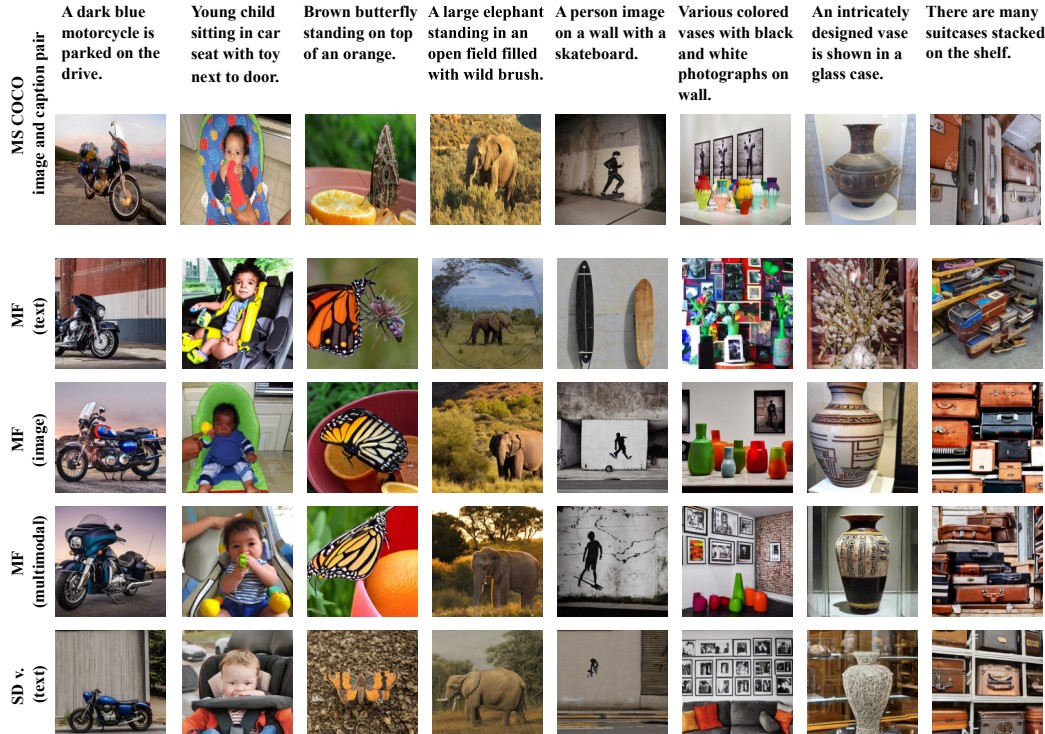

Figure 7: Qualtiative examples of images generated from MS COCO images and captions. We compare outputs from MULTIFUSION (MF), prompted with different modalities - text, image, and multimodal - as well as outputs from Stable Diffusion, which was prompted with text only. We use guidance scale 4.0 for both models.

# A    Experimental protocol

Subsequently, we provide further details on the datasets (4), architecture and training details (3) of MULTIFUSION. We start from Stable Diffusion v1.4 and discard the CLIP text encoder. MULTIFUSION's encoder is built on a 13B decoder-only transformer similar to GPT-3 [10] but using rotary position embeddings [43]. This LM was pre-trained on a multilingual corpus of roughly 400B tokens comprised of English, German, French, Italian, and Spanish data (cf 4).

We turned the model into a visual language model (VLM) following the method of MAGMA [15]. Therefore, we used a ResNet image encoder, initialized with CLIP weights, and extract the feature grid before the final pooling layers. These 12x12 features are then flattened to a sequence of 144 vectors with subsequent drop regularization and layer normalization resulting in the embedding representation of an image. Consequently, an image is represented by 144 token embeddings which are used as inputs to the transformer LM alongside the text embeddings. For parameter-efficient multimodal training, we added a bottleneck adapter with downsampling factor 8 [23] to each attention and feed-forward layer. Only the weights of these adapters and the image prefix module were optimized during multimodal pre-training. The multimodal components (image prefix and adapters) are trained autoregressively for 15.4 million image-text pairs on a proprietary multimodal dataset (cf 4). Following MAGMA [15], the image tokens are prepended to the text tokens, and the language modeling loss for next token prediction is computed over the text only. We based our multimodal dataset on the findings and extensive ablations of MAGMA [15].

In addition to the tuned multimodal adapters, we apply parameter-efficient bias finetuning [4] of the pre-trained language model to improve the semantic alignment of encoded representations. This turns the hidden representations of the transformer, learned during pre-training, into semantic embeddings usable for arbitrary downstream tasks. Our setup is similar to S-GPT [31], pooling the last hidden layer of the decoder into dense representations, which we optimized for semantic search. We used a

Table 3: Training details of MF's individual modules.

| Module | Parameters | Trained parameters (MF) | Training data size | GPU hours |
|---|---|---|---|---|
| LM | 13B | - | 400B tokens | 85K |
| Semantic embeddings | - | - | 50M tokens | 6.9K |
| Multimodal components | 2B | - | 15.4M image-text pairs | 9.5K |
| SD (v1.4) | 1B | 152M (15%) | 40M images | 4.3K |

Table 4: Training datasets of MF's individual modules. Only the semantic bias training uses paired multi-lingual data. All other training data is sourced from monolingual corpora.

| Module | Dataset | Percentage | Languages | Modalities |
|---|---|---|---|---|
| LM | web crawl | 71% | english, german, italian, spanish, french | text |
| | books | 20% | | |
| | political and legal sources | 5% | | |
| | wikipedia | 2% | | |
| | news | 2% | | |
| | other | 1% | | |
| Semantic embeddings | MNLI | 72% | english, german | text |
| | SNLI | 28% | | |
| Multimodal components | image captioning tasks | 91% | english | text, image |
| | Wikipedia image-text | 8.5% | | |
| | visual question-answering tasks | 0.5% | | |
| SD (v1.4) | LAION aesthetics | 100% | english | text, image |

custom version of SNLI [7] and MNLI [47] that extends the original English texts by their machine-translated German versions, which were generated using the DeepL API. Both datasets contain natural language tuples of premise and hypothesis along a judgment label that can either be entailment, contradiction, or neutral. For example, the hypothesis '*The man is sleeping*' is a contradiction to the premise '*A man inspects the uniform of a figure in some East Asian country.*' We optimize the bias weights of the LM using a contrastive learning objective for 13k steps where entailments serve as positive samples and contradictions as negative ones.

The diffusion model itself is kept frozen with only the cross-attention layers of the U-Net being re-trained to utilize MULTIFUSION's embedding space. As training data, we use LAION aesthetics V.2 5+[8], i.e., the subset of LAION 5B with English captions filtered by a predicted aesthetic score $> 5$ [42]. Additionally, we discard images with a resolution lower than $512 \times 512$, resulting in roughly 40M captioned images. The final model is finetuned for 60k steps with the probability of using an image instead of a caption being $0.2$.

We make some modifications to the mechanics at inference to better enable multimodal prompting. For interleaved text and image prompts, MULTIFUSION's encoder represents images as a sequence of 144 token embeddings. This results in a disproportional influence on the generated image compared to significantly shorter text portions. To counteract this phenomenon, we utilize attention manipulation [14] in every attention layer of the transformer encoder. Meaning that every attention score $s_{ij}$ in an attention head is modified by a per-token factor $\lambda_i$ so that $\tilde{s}_{ij} = s_{ij} + \log \lambda_i$. The authors argue, that $\lambda_i > 1$ up-weights the $i$-th token, whereas $\lambda_i < 1$ down-weights it. In practice, we use the same $\lambda$ value to up-weight each text token of a prompt.

# B   MS COCO examples

In Fig.7, we show exemplary images generated from MS COCO prompts in all modalities (cf. Sec. 4). We show that multi-modal prompts enhance image quality as well as convey more information while still producing diverse and meaningful outputs, which are not mere copies of the original input image.

---

[8]https://laion.ai/blog/laion-aesthetics/

# C MCC-250 – user study

Subsequently, we present further implementation details of the user study conducted on the MCC-250 benchmark (cf. Sec. 4), along with qualitative examples.

## C.1 Study Details

For each model, we generated 10 images per prompt. Let's consider the example '*a green bench and a red car*' from the benchmark. The prompt to Stable Diffusion was '*a photorealistic image of a green bench and a red car*', whereas we prompted Composable Diffusion with the two prompts '*a photorealistic image of a green bench*' and '*a photorealistic image of a red car*' which were composed with equal weights. For MULTIFUSION, we interleaved the prompt with image references of each object: '*a photorealistic image of a green bench <image of a green bench> and a red car <image of a red car>*'.

We evaluated each of the two objects and the respective attribute separately, with questions being posed in the form:

> Is there a green bench in the image?

The four answer options were:

- There is no bench
- There is a bench, but it is not green
- There is a green bench
- Cannot tell

Each user was tasked with labeling a batch of 28 image/attribute pairs; 25 out of those were randomly sampled from our generated images. Each batch contained three hand-selected images from the MCC-250 inputs as a sanity check. If users labeled these three images incorrectly, the batch was discarded and added back to the task pool. Each image/object combination was labeled by three different annotators resulting in annotator consensus if at least 2 selected the same label. The results in the main paper (cf. Tab. 2) only include those images with annotator consensus, which are distributed according to Tab. 5.

Table 5: Annotator consensus on the selected label per model for the MCC-250 user study.

| Model | Annotator Consensus in [%] |
|---|---|
| Stable Diffusion | 98.31 |
| Composable Diffusion | 93.85 |
| MULTIFUSION (text) | 98.71 |
| MULTIFUSION (multimodal) | 98.16 |

To conduct our study, we relied on Amazon Mechanical Turk, where we set the following qualification requirements for our users: HIT Approval Rate over 95% and at least 1000 HITs approved. Annotators were fairly compensated according to Amazon MTurk guidelines. Users were paid $0.60 for a batch of 28 images.

## C.2 Qualitative Examples

We show further examples of images generated on the MCC-250 benchmark in Fig. 8. The results in Fig. 8a further highlight the robustness of MULTIFUSION for image composition tasks. When prompted with multimodal inputs, the model reliably includes both objects with correct attributes in the generated image. In contrast, Stable Diffusion oftentimes fails to render one of the objects, and Composable Diffusion mostly generates strange mixtures of objects where none of them can be clearly made out.

However, we observed MULTIFUSION to perform consistently poorly on prompts where both objects were animals. As shown in Fig. 8b the generated images contain creatures that exhibit features

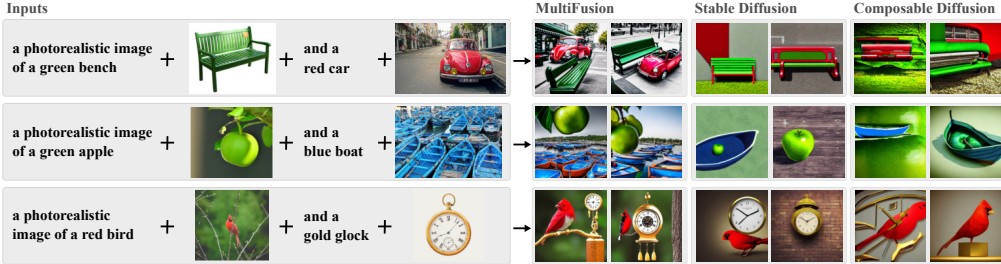

(a) Positive examples highlighting the improvement of MULTIFUSION over baseline models.

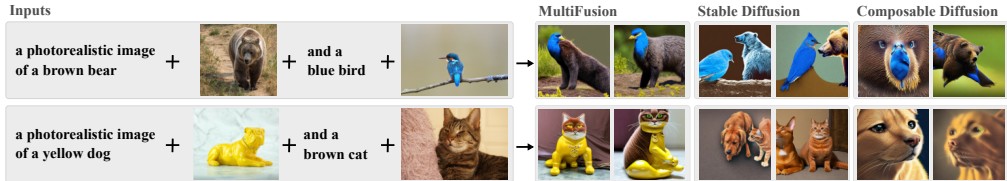

(b) Failure cases of MULTIFUSION. Specifically, for two animals the model often fails to produce two distinct objects and instead generates mixtures of both.

Figure 8: Representative examples of generated images on the MCC-250 benchmark. MULTIFUSION is prompted with interleaved multimodal inputs, whereas Stable and Composable Diffusion are prompted with text only.

Table 6: Fine-grained human evaluation results on **animal compositions**. Showing results for the subset of MCC-250 where both are animals. SD and Composable Diffusion were prompted with text, whereas we used multimodal prompts for MULTIFUSION, containing one interleaved visual reference for each object. Recall that each prompt is a complex conjunction of two different animals with different colors.

| Methods | Zero obj ↓ | One obj. ↑ | One obj. w/ correct color ↑ | Two obj. ↑ | Two obj. w/ correct colors ↑ |
|---|---|---|---|---|---|
| Stable Diffusion [%] | $\mathbf{0.00}_{\pm 0.00}$ | $\mathbf{100.00}_{\pm 0.00}$ | $\mathbf{87.24}_{\pm 19.00}$ | $\mathbf{41.51}_{\pm 27.04}$ | $\mathbf{31.99}_{\pm 27.96}$ |
| Composable Diffusion [%] | $4.15_{\pm 7.90}$ | $95.67_{\pm 8.18}$ | $81.13_{\pm 18.29}$ | $18.41_{\pm 19.04}$ | $9.46_{\pm 10.95}$ |
| MultiFusion (text) [%] | $2.57_{\pm 6.92}$ | $97.43_{\pm 6.92}$ | $83.60_{\pm 20.31}$ | $26.22_{\pm 22.95}$ | $21.38_{\pm 22.70}$ |
| MultiFusion (multimodal) [%] | $3.56_{\pm 7.36}$ | $96.45_{\pm 7.36}$ | $83.60_{\pm 20.31}$ | $26.22_{\pm 22.95}$ | $21.38_{\pm 22.70}$ |

of both animals. For Stable Diffusion, on the other hand, there is no significant difference in performance on this specific sub-task. In Tab. 6, we show the results of the human evaluation on the animal-only subset. While Stable Diffusion performs comparably to the entire benchmark, both MULTIFUSION and Composable Diffusion portion of correctly generated images drops by over 50%.

## D   Quantitative and qualitative study of attention manipulation

To evaluate the effect of attention manipulation we compute additional FID scores for the multimodal prompt ablating the attention manipulation weight on the text prompt. In Tab 7 we compare the different FID scores of the multimodal prompts to those of image and text prompts and observe that the higher the weight the closer the FID score is to a text-only prompt. This quantitatively verifies that a higher attention manipulation weight on text prompt increases its influence on the generated image. Further, we qualitativley demonstrate the use of attention manipulation [14] in combination with multi-modal prompts in Fig. 9. We consistently increased the weight of the text prompt and illustrate how this correlates with an increasing influence of the text prompt on the generated images. If the text is weighted equally to the input image, its effect on the generated output remains negligible. Attention manipulation at the input encoder offers an intuitive interface to balance the influence of each input modality. In general, we observed that text prompts should be up-weighted by a factor of 10-25 to yield the desired results. Additionally, the manipulation behaves quite robustly with no adverse effects on the generated images for higher weights. Further, we show the interpolation

Table 7: Ablation of attention manipulation with FID-30k on the MS-COCO validation set. The weight of the text prompt is increased from 1 to 20.

| Guidance Scale | COCO FID-30K ↓ | | | | |
|---|---|---|---|---|---|
| | Image | Multimodal (1:1) | Multimodal (10:1) | Multimodal (20:1) | Text |
| 4.0 | 6.03 | 6.74 | 8.53 | 9.22 | 9.90 |

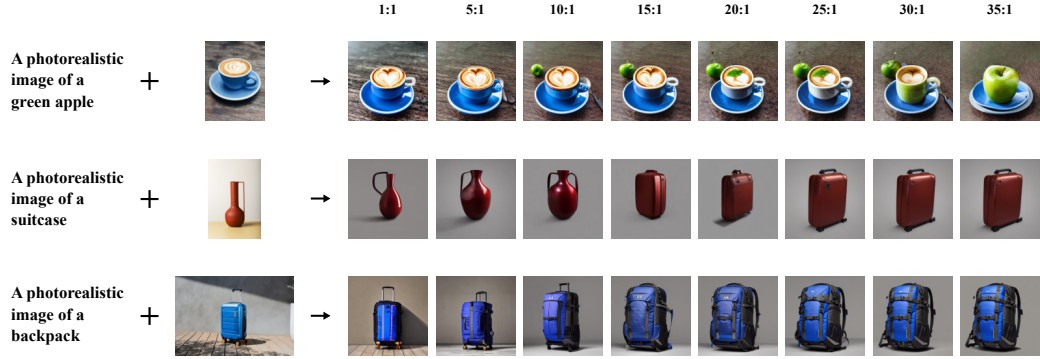

Figure 9: Qualitative examples demonstrating the use of attention manipulation [14] in combination with multimodal prompts and its effect on the generated images. We increase the weight on the text prompt from left to right with the exact weights being depicted in the top line as *text_weight : image_weight*.

between two objects - one represented by a text and one by an image prompt - through the use of attention manipulation in Fig. 10. Here we can also observe that text prompts up-weighted by a factor of 10-25 result either in display of both or a shift of objects.

Figure 10: Qualitative ablation of attention manipulation [14] in combination with multimodal prompts of two objects and its effect on the generated images. The text prompt weight is increased from 1 to 35, with the exact weights being depicted in the top line as *text_weight : image_weight*.

# E   Qualitative examples of input order

The interleaved inputs are concatenated to one input vector and subsequently fed into the LM, which outputs embeddings for conditioning MultiFusion's image generation U-Net. Changing the order of interleaved inputs will change the embedding produced by the encoder and thus affect the conditioning of the denoising process, leading to a different output. This can be attributed to the autoregressive generation of embeddings with causal attention by the LM. The effect is particularly important for image prompts, where the relationship between multiple concepts is not specified in natural language. Thus, we provide qualitative examples of reversed image prompts in Fig. 11. We can observe the

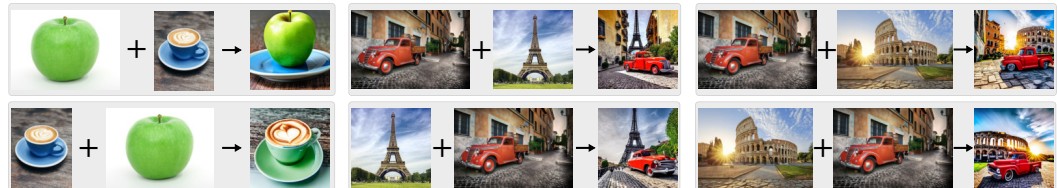

Figure 11: Qualitative examples illustrating the effect of reversing the input order of prompt images.

effects of autoregression and causal attention in all three examples of Fig. 11, showcasing that the object or background of the first input image has the highest influence on the output image.

# F  Further Qualitative Examples

Finally, we provide further qualitative examples, which showcase the multi-modal and multilingual capabilities of MULTIFUSION in Fig. 12. These images further highlight the overall versatility and expressiveness of the model.

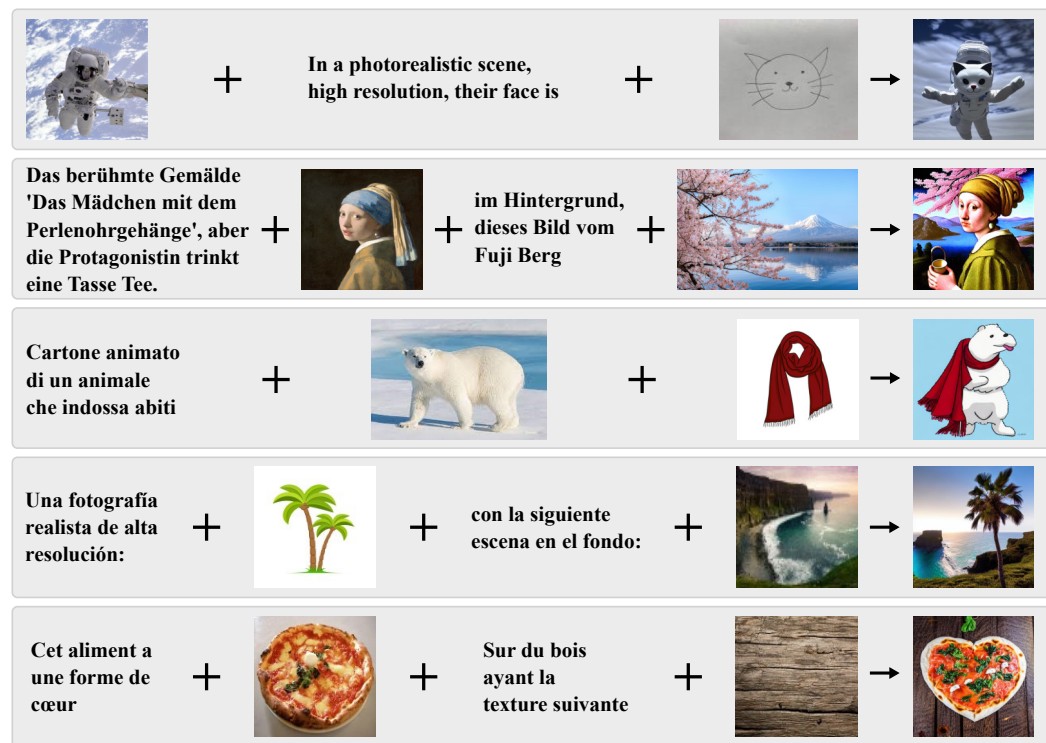

Figure 12: Further qualitative examples of interleaved multimodal prompting. We provide one example in each language supported by MULTIFUSION: English, German, Italian, Spanish and French.

