# OpenReview forum: "MultiFusion: Fusing Pre-Trained Models for Multi-Lingual, Multi-Modal Image Generation"
_NeurIPS.cc/2023/Conference — NeurIPS 2023 poster_

### Official Review · Reviewer_2Kwy · 2023-07-05

**Soundness:** 4 excellent
**Presentation:** 3 good
**Contribution:** 4 excellent
**Rating:** 7
**Confidence:** 3

**Summary:**

This paper presents MultiFusion, a multilingual multimodal image generation model, which can be effectively trained by fusing existing pre-trained visual models, language models, and stable diffusion models. Using a multilingual autoregressive language model as a bridge, MultiFusion follows MAGMA to enable multimodality by learning adapters. Before connecting the language model with the stable diffusion module, it learns semantic embeddings with a contrastive learning objective in a parameter-efficient setup. Finally, it connects the language model with the diffusion model with monomodal data. i.e., an image or caption.

Experimental results demonstrate that the trained MultiFusion model can generate high-quality images with multimodal interleaved prompts. Besides, with modular design and the fusion of pre-trained models, the training can be quite effective compared to training from scratch. Several analyses such as attention manipulation also provide insights into the multimodal language models.

**Strengths:**

- The paper makes a clever combination of pre-trained models and adapter learning techniques, including 1) MAGMA for cross-modal adaptation/fusing, 2) contrastive learning before fusing LM with the diffusion module, 3) cross-attention learning of SD to align the conditioning with the new embedding space. These operations delicately combine the pre-trained modules into an end-to-end multimodal-text-to-image model.

- Experimental results show that MultiFusion can produce high-quality images conditioned on multimodal and multilingual inputs, with a wide range of applications and use cases.

- The analysis on attention manipulation is quite interesting.

**Weaknesses:**

- It would be great to improve the presentation of the paper, especially methods and implementation details. Although Figure 1 presents an overview of the architecture, I have to guess some of the implementation details and carefully find clues from a large amount of text. Some suggestions: (1) you could provide some figures to show the details of how the adapters connect the pre-trained models and how you learn them; (2) you could also clarify the training tasks, and data in tables.

- Existing works have explored how to learn adapters to connect pre-trained modules. For example, Flamingo learns gated adapter modules to connect language models with visual models, and generate text conditioned on multimodal inputs. The paper provides a full solution to the problem with careful design but it is kind of an integration of existing adapter methods.

**Questions:**

- Are all the outputs of the language model connected to the diffusion model or only the last token?

E.g., Input: [img_tok1] [img_tok2] [text_tok1] [text_tok2] [text_tok3] -> output vectors: [h1] [h2] [h3] [h4] [h5].

Are all five vectors passed to the diffusion model or only the last one?

- Why is an autoregressive model used as the encoder? I understand the CLIP model has disjoint text and image encodings, but why not use bidirectional models as the multimodal encoder? Any explanations?

**Limitations:**

As mentioned in the paper, the model always produces variations of input images, which limits its applications in image editing. I think it is worth mentioning this limitation, which provides further understanding of MultiFusion.

---

> ### Author Rebuttal · Authors · 2023-08-10
>
> We thank reviewer 2Kwy for their feedback and suggestions. Below we address each weakness and question separately. Tab. 5-7, as well as Fig. 11 and 12, can be found in the supplementary PDF of this review.
> ### W1) Architectural details
> We thank the reviewer for the suggestions and incorporated this feedback in our global response regarding system design. The adapter layers are added to each attention and feed-forward layer of the transformer. Following the method proposed by MAGMA [16], the adapters are trained autoregressively on a combination of large scale image-text datasets (c.f. Tab.5 and 6). The image tokens are prepended to the text tokens and the language modeling loss for next token prediction is computed over the text only. We adjusted Fig. 1 and its caption to clarify these details.
> Additionally, we supply further information on the training data sizes, parameter counts, and GPU hours for all components in Tab. 5, along with details on data splits, languages, and modalities in Tab 6. We will add both tables to the final version of the paper.
>
> ### W2) Pre-existing work on adapters
> We agree with the reviewer’s assessment that MultiFusion builds on previous work for multimodal adapter tuning. Nonetheless, we want to highlight that MultiFusion’s contribution lies in the investigation of building a cohesive system for a complex downstream task utilizing these pre-trained components, thus introducing a novel method of expressing prompts and steering image generation for diffusion models, and significantly reducing the computational costs required for this system.
> ### Q1) Embedding tokens
> Indeed, the hidden representations of all tokens are passed to the diffusion model. In the example outlined by the reviewer, this would mean 5 embedding vectors are used for image generation conditioning.
> ### Q2) Autoregressive encoder
> We argue that decoder models perform better on tasks such as manipulation with natural language (“Subject X with background Y and style Z”) (cf. qualitative results) or correct features attributions (“Red X, Blue Y”) (cf. MCC-250 results). These capabilities can be attributed to the natural breaking of permutation equivariance [54], compared to bidirectional models relying entirely on positional embeddings.
> Furthermore, as demonstrated in Flamingo [55], multi-modal decoders can reason over multiple images, enabling further flexibility in the model’s capabilities (e.g., elements from multiple images can be trivially composed together, as demonstrated in Fig.2 and Fig.5).
> We acknowledge that bi-directional models may outperform autoregressive ones on other embedding tasks [56], but argue that an autoregressive model is better suited for the tasks studied in MultiFusion due to the benefits outlined above.
>
> [16] Constantin Eichenberg, Sidney Black, Samuel Weinbach, Letitia Parcalabescu, Anette Frank.
> MAGMA - Multimodal Augmentation of Generative Models through Adapter-based Finetuning. EMNLP, 2022.
> [54] Amirhossein Kazemnejad, Inkit Padhi, Karthikeyan Natesan Ramamurthy, Payel Das, and Siva Reddy. The Impact of Positional Encoding on Length Generalization in Transformers. arXiv preprint arXiv:2305.19466, 2023.
> [55] Jean-Baptiste Alayrac, Jeff Donahue, Pauline Luc, Antoine Miech, Iain Barr, Yana Hasson, Karel Lenc et al. Flamingo: a visual language model for few-shot learning. In Proceedings of the Conference on Neural Information Processing Systems (NeurIPS), 2022.
> [56] Michihiro Yasunaga, Armen Aghajanyan, Weijia Shi, Richard James, Jure Leskovec, Percy Liang, Mike Lewis, Luke Zettlemoyer, and Wen-tau Yih. Retrieval-augmented multimodal language modeling. arXiv preprint arXiv:2211.12561, 2022.

---

> > ### Comment · Reviewer_2Kwy · 2023-08-20
> >
> > Concerns are addressed. Thank you. The overall score is updated.

---

### Official Review · Reviewer_5v9N · 2023-07-05

**Soundness:** 3 good
**Presentation:** 3 good
**Contribution:** 3 good
**Rating:** 7
**Confidence:** 4

**Summary:**

In this paper, the authors present a novel approach to expressing complex concepts with arbitrarily interleaved multimodal and multilingual input. Their approach leverages pre-trained models and allows an efficient fusion of different component without training a model from scratch.

**Strengths:**

1. The paper is well-written and easy to follow
2. The experiments are well-designed and allow one to use existing pre-trained models while reducing the demand to train a system from scratch.
3. The result on various benchmarks are promising and would invite more discussion in this line of work.

**Weaknesses:**

The motivation to attempt such a problem is rather weak. Under what circumstances, would one want to have interleaved multimodal input to generate images? Is it because we want to control the input? If so, why not compare the proposed approach with similar models such as ControlNet and DreamBooth?

**Questions:**

See my comments in Weaknesses

---

> ### Author Rebuttal · Authors · 2023-08-09
>
> We thank the reviewer for their feedback and subsequently explain the motivation of MultiFusion in more detail.
> ### W1) Motivation and comparison.
> **Motivation**
> There exist several motivations for using interleaved multimodal inputs. As demonstrated by the experiments in the paper, reference images often contain more fine-grained and detailed information than textual descriptions can provide (for example, for the style of an image as in Fig. 6b). Furthermore, visual inputs (next to textual) are universally understood by users further broadening model accessibility. On the other hand, text offers more abstract control of the generation and is able to express complex concepts. Consequently, combining the two modalities offers the best of both worlds, substantially increasing model expressibility.
>
> **ControlNet and Dreambooth**
> Dreambooth, on the other hand, particularly focuses on generating specific subjects rather than using images as arbitrary reference information. Furthermore, DreamBooth requires computationally expensive fine-tuning for each subject, whereas MultiFusion can use image inputs natively during inference – again increasing accessibility.
> Similarly, ControlNet focuses on a completely different problem than Multifusion in providing dedicated control over scene composition using low resolution inputs such as edge maps. Moreover, ControlNet requires training a dedicated model for each control modality.
> In fact, we believe DreamBooth and ControlNet to be orthogonal approaches and see use cases where either may be used in combination with MultiFusion.

---

### Official Review · Reviewer_g96C · 2023-07-07

**Soundness:** 1 poor
**Presentation:** 1 poor
**Contribution:** 2 fair
**Rating:** 4
**Confidence:** 4

**Summary:**

In this paper, the authors introduce MultiFusion, a novel approach that enables the expression of complex and nuanced concepts in text-to-image diffusion models (DM) through arbitrarily interleaved inputs of multiple modalities and languages. The “fusion” concept is at the core of the whole work: to fuse modalities together, pre-trained models (a LLM and a stable diffusion backbone) are fused together. Experimental results highlight the efficient transfer of capabilities from individual modules to the downstream image generation module. Notably, MultiFusion empowers the image generation module to effectively utilize multilingual, interleaved multimodal inputs, even when trained solely on monomodal data in a single language. The contributions of this work include the fusion of modalities for image generation, experimental evaluations, and the introduction of a benchmark dataset for further analysis and comparison regarding the multimodal compositionality of the models.

**Strengths:**

1. **Innovative model fusion approach**: The paper introduces an innovative approach by combining a partially frozen multilingual Language Model (LLM) with a stable diffusion backbone. This fusion results in an interesting multilingual and multimodal encoder capable of seamlessly interleaving between input items, treating them as a modality-agnostic sequence.
2. **Multilingual alignment investigation**: The authors conduct an investigation into the model's multilingual capabilities by translating the prompts from the DrawBench dataset. This exploration demonstrates an understanding of the importance of multilingual alignment. While there is a question regarding the accuracy of the translations, the authors acknowledge the potential benefit of utilizing literal translations in training the multilingual encoder, even though nuances in meaning may not be fully captured. This highlights the authors' attention to addressing the challenges and complexities of multilingual representation.
3. **Contribution of benchmark dataset**: The authors contribute to advancing research in multimodal compositionality by producing and sharing the MCC-250 dataset. This benchmark dataset, described in detail in the supplementary material, serves as a valuable resource for assessing the compositionality of multimodal inputs, specifically comprising English text and images. The production and release of this dataset demonstrate the authors' dedication to promoting reproducibility, comparison, and further progress in the field of multimodal compositionality.

**Weaknesses:**

1. **Lack of clear architectural design and novelty**: The paper suffers from a lack of clarity in explaining and justifying its design choices. While references are provided, the underlying motivations and problem-solving aspects of these choices are not adequately explained. While Figure 1 attempts to illustrate the model structure, it is not accompanied by a clear rationale and explanation for the chosen modules and their interactions in the text. Enhancing the clarity of the architectural design would elevate the novelty and originality of the proposed approach. It is suggested to provide a high-level description that guides the reader in understanding the motivations behind specific choices. By focusing on the "why" rather than the low-level details, readers can, for example, grasp the purpose of unlocking only the biases in the LLM. Currently (line 130-131), it is unclear if this is a crucial step to obtain good results while keeping a parameter-efficient regime, or if it is marginal in that regard. The supplementary material can be utilized to provide additional low-level details for interested readers.
2. **Lack of clarity in Figure 4 and semantic search paragraph**: While Figure 4a demonstrates higher similarities of translated prompts in the authors' method compared to competitors, it does not provide insights into the similarities between the reference and other negative samples. This additional information is crucial to establish the range of similarities that can be considered as genuinely low. Furthermore, Figure 4b indicates that the AltDiff competitor generates potentially more consistent images in each language, suggesting that the embedding similarity between references and translations may not be entirely representative. Clarifying these aspects would enhance the understanding of the results and provide a more comprehensive evaluation of the proposed method's performance.
3. **Missing standard deviation in tables**: Including standard deviation in the results would provide important information about the variability and statistical robustness of the findings. By incorporating this measure, the paper would strengthen the reliability and credibility of the reported results.
4. **Performance comparison and insights**: In Figure 4b, the AltDiff method demonstrates better performance, raising questions about the potential benefits of adding more languages to the MultiFusion method. While the authors suggest that alignment remains similar despite MultiFusion being fine-tuned using only English data, additional experiments are needed to provide substantial evidence that adding more languages to MultiFusion indeed yields improved results. Further investigation and insights in this area would enhance the value and understanding of the proposed method.

In offering these critical observations, I would like to emphasize that my intention is not to be harsh, but rather to provide constructive feedback. I acknowledge that explaining such a complex pipeline can be challenging and that significant effort has been invested in this work. However, I strongly believe that there is room for improvement in describing the architectural choices and highlighting the strengths of the paper, and I’ve tried my best to give possible suggestions in this regard. I’m convinced that addressing these aspects would significantly improve the quality and impact of the paper, but I don’t think this is something that could be fixed within the rebuttal period. In any case, I remain open to reconsidering my recommendation if any relevant insights emerge during the discussion.

**Questions:**

**Lack of Chinese column in Figures 4a and 4b**: It is unclear why this column is missing or was included solely for the competitors if the proposed method was not tested on this language.

There's a typo on line 139: extracted

**Limitations:**

The authors adequately addressed the limitations.

---

> ### Author Rebuttal · Authors · 2023-08-10
>
> We thank the reviewer for their constructive feedback. Tab. 5-7 as well as Fig. 11 and 12 can be found in the supplementary PDF of the rebuttal.
> We agree on the importance of design justification. We reiterate on design choice details for clarity in our global response under the section "Reiteration on system design".
>
> ### W1) Architectural design
> Previous work has demonstrated context-sensitive LM text encoders improve the expressiveness of downstream image generation models [40,2]. Accordingly, we model the backbone of MultiFusion’s (Mf) encoder as a 13B autoregressive transformer [10] trained on a multilingual corpus. We justify this decision over a bi-directional architecture by arguing that decoder models outperform bi-directional models on tasks such as manipulation with natural language (“Subject X with background Y) (cf qualitative results) or correct features attributions (“Red X, Blue Y”) (cf MCC-250 results). These capabilities have previously been attributed to the natural breaking of permutation equivariance [54], compared to bidirectional models relying entirely on positional embeddings.
> Following MAGMA [16], we add an image prefix and dedicated adapters to enable multimodal capabilities. We argue that adapters are a suitable architectural choice for multimodal prompts, as previous research has already performed extensive ablations on adapter architectures and demonstrated their improved understanding of multimodal inputs over other methods [16].
> Our choice of semantic embeddings was guided by the intuition that a focus on the semantics of a text prompt would best capture the information relevant to image generation and thus simplify learning the mapping from embeddings to image outputs. We decided to obtain high-quality semantic embeddings through parameter-efficient bias [4] instead of full model finetuning, based S-GPT [33].
> Early experiments have confirmed higher rates of convergence (based on visual inspection of generated outputs) for experiments using semantic embeddings. Consequently, bias tuning is an essential condition and not an optional architecture choice for successfully fusing an image generation model.
> In line with previous research [13], we finetune the cross-attention parameters of SD on LAION aesthetics.
> We adjusted Fig. 1 & caption and section 3 to clarify details on the architecture and to better reflect the design choices. We would like to clarify that the expected level of implementation details in the main body is highly subjective. We aim to strike a balance between high-level motivation and low level details to satisfy the majority of readers. Indeed, multiple of the other reviewers specifically asked for more low-level information in the main body of the paper.
> [54] arXiv:2305.19466
>
> ### W2) Fig 4a and semantic search + W4) Comparison and insight
> We agree that the addition of a baseline similarity between uncorrelated sentences is useful for the interpretation of Fig 4a. In fact, the baseline is roughly equivalent to the performance of CLIP reported on zh prompts, suggesting that the regular CLIP model is not aligned for this language and further reinforcing the increased performance of MultiFusion. We explicitly marked the baseline in Fig 4a).
> We believe there to be a misunderstanding in the assessment of Fig 4b, which we would like to address. We do not believe that the results allow for a strong statement of AltDiffusion (AD) outperforming MF or vice versa due to the significant overlap in error bars. In fact, a good alignment of AD’s images is to be expected as the model’s image generative training is explicitly done on (aligned) multilingual data. The key insight from this experiment lies MF achieving comparable performance despite the image generation being only trained on English data. We attribute this capability to better-aligned embeddings. These alignment results suggest that aligned multilingual data on a downstream task is not necessary to achieve alignment. Rather, good embedding alignment of the backbone model in combination with readily available monolingual task-specific data is sufficient for multilingual alignment on that task.
>
> Investigating more languages is indeed an interesting avenue for future work. However, the error bars already indicate that the performance of AD is not significantly better than MF’s. Thus demonstrating MF’s potential benefit in low-resource domains with only a few or even no image-text pairs available. This is, however, out of the scope of the current contribution and, therefore, also not a claim we make in the paper.
>
> ### W3) Standard deviations
> We agree that standard deviations (std) generally improve the interpretability of results, wherefore we included error bars in Fig. 4.
> In line with the literature do not report std for FID and CLIP in Table 1.
> We did not initially include std for the empirical analysis in Tab. 2 as we simply reported the binary success rate over the entire benchmark. By considering the per prompt success rate over multiple samples, we now also report standard deviations over differing object compositions and will extend Tab. 2 accordingly. All models exhibit comparatively high stds (ca. 30PP for 2 objects), suggesting a fair amount of outliers for which the models perform significantly better/worse than on average. We believe the investigation of these tasks to be a promising avenue for future work and adjusted the discussion of the results accordingly.
> ### Q1) Lack of Chinese columns
> The models presented in Fig. 4 were evaluated on the languages that they have been trained on. Thus, we do not provide scores for AD on German and MF on Chinese. Further, we argue that it is crucial to include scores for MF on German and AD on Chinese, as these are the respective secondary languages of the models, i.e. the ones with the most aligned training data. We adjusted the caption of Fig. 4 to better reflect this specific aspect of the experiment.

---

> > ### Comment · Reviewer_g96C · 2023-08-21
> >
> > I appreciate the author's response, and I thank them for that.
> > While I remain somewhat unconvinced by the paper's contribution and robustness, considering the perspectives of other reviewers and the thorough response, I am inclined to adjust my rating to a borderline reject (4).

---

### Official Review · Reviewer_sjMA · 2023-07-08

**Soundness:** 3 good
**Presentation:** 3 good
**Contribution:** 4 excellent
**Rating:** 7
**Confidence:** 3

**Summary:**

This work proposed a novel method to build multilingual multimodal generation models that supports prompts composed of interleaved text and image. It combines a strong pre-trained multilingual language model with the image generation model from Stable Diffusion (SD) and achieves alleviated capabilities such as prompting with text and image combined. It also shows that the new model does better in composition generations, as one can provide reference images as part of the prompt.

In this work, a multilingual language model is trained in the first place (13B encoder-decoder structure model trained on 400B tokens), which itself is a strong multilingual model. It then adds an adapter module to the LLM model to support input in image format, following methods proposed in MAGMA. Finally, it aligns the trained encoder with the diffusion model taken from Stable Diffusion, with 15M text-image pairs. As a result, it can support multilingual, multimodal prompt for image generation, without training on massive text-image pairs dataset.

In experimentation, it showed that using both text and image as prompts can be beneficial, especially in composition generations. It also shows superior performance to existing multilingual text-image generation model AltDiffusion, possibly due to better alignment in multilingual embeddings. Further, the support of taking image as prompts can enable varies applications such as negative prompting with image, image composition, image variation and style modification.

**Strengths:**

* Novel and efficient method: fusing different pre-trained models works very well which can bootstrapping existing models such as stable diffusion to achieve different input format, avoiding the heavy cost of training model from scratch.
* Enables prompting using both image and text and generates better images both in terms of metrics such as FID and human evaluation,  comparing to baselines that only takes text prompt.
* Better results on composition generations from considering reference images in prompts.
* Well written overall and addressed limitations of the work very well.

**Weaknesses:**

* Some of the details such as model parameter size, training data source and size are not presented in the main paper (included in appendix), which can be less clear when interpreting results presented in experimental section. It would be better to point those factors out when comparing with baselines in the main paper.



**Questions:**

1. As you mentioned in the limitation section, the model cannot produce images that’s exact or close to the image in prompt. I wonder if there can be simple modifications made that can achieve this?
2. How is the interleaved data being used? Does it effect the results if you change the order of the interleaved data?
3. Have you tried other methods in addition to adapter to fuse image and text modality?
4. Have you done any ablation studies on the semantic embeddings?
5. On the MSCOCO dataset for generation, where is the image prompt come from (referring results in table 1)?
6. Have you compared the multilingual LM trained to other ones (mT5 etc.) in multilingual benchmarks?


**Limitations:**

The authors addressed the limitations relatively well in the paper:
1. The generated image cannot do copy of exact prompt images;
2. Sometimes the image prompts need to be carefully chosen and do not always work;
3. It suffers the same shortcomings (such as inappropriate content) as other generation models trained on very large-scale crawled dataset (LAION).

---

> ### Author Rebuttal · Authors · 2023-08-10
>
> We thank the reviewer for their feedback and subsequently address each limitation and question separately. [Tab. 5-7, as well as Fig. 11 and 12, can be found in the supplementary PDF of this review]
>
> ### W1) Training details
> We supply further information on the training data sizes, parameter counts, and GPU hours for all components in Tab. 5 along with details on data splits, languages and modalities in Tab 6. We will add both tables to the final version of the paper.
>
> ### Q1) Image replication from prompt
> Direct replication of an input image would most likely require a different architecture used in encoding embeddings. While this in itself is an interesting research question, our approach aims to enable a more fine-grained conditioning of the image generation process through multimodal prompts that can be arbitrarily interleaved with one or more images. Nonetheless, our diffusion model can easily be combined with existing diffusion-based image editing techniques that can faithfully reconstruct and subsequently alter an image [58, 59]. In this case, MultiFusion would facilitate multimodal image editing, which we believe to be a promising avenue for future research.
>
>
>
> ### Q2) Interleaved input data and order influence
> The interleaved inputs are concatenated to one input vector and subsequently fed into the LM, which outputs embeddings for conditioning MultiFusion’s image generation U-Net. Changing the order of interleaved inputs will change the embedding produced by the encoder and thus affect the conditioning of the denoising process, leading to a different output. This can be attributed to the autoregressive generation of embeddings with causal attention by the LM. The effect is particularly important for image prompts, where the relationship between multiple concepts is not specified in natural language. Thus, we provide qualitative examples of reversed image prompts in Fig. 12, which we add to the appendix. We can observe the effects of autoregression and causal attention in all three examples of Fig. 12, showcasing that the object or background of the first input image has the highest influence on the output image.
>
> ### Q3) Ablations on multimodal architecture
> We limited our investigation to adapters for multimodal fusion. Previous research has already performed extensive ablations on adapter architectures and demonstrated their improved understanding of multimodal inputs over other methods [16].
> Based on these findings, we argue that adapters are a suitable architectural choice for the task at hand. We adjusted Section 3 accordingly to better reflect these design choices.
>
> ### Q4) Ablations on semantic embeddings
> Our choice of semantic embeddings was guided by the intuition that a focus on the semantics of a text prompt would best capture the information relevant for image generation and thus simplify learning the mapping from embeddings to image outputs. We decided to obtain high-quality semantic embeddings through parameter-efficient bias [4] instead of full model finetuning, based on the work of [33]. Early experiments have confirmed higher rates of convergence (based on visual inspection of generated outputs) for experiments using semantic embeddings. Consequently, we do not report ablation results without semantic fine-tuning, as it is an essential condition and not an optional architecture choice for successfully fusing an image generation model. We adjusted section 3 of the paper accordingly to provide more clarity on this behavior.
>
>
>
> Q5) MS-COCO image prompt
> The reference image used for the experiment in Tab. 1 is the ground truth image from COCO. We realize this is strong supervision, most likely not available for a real-world use case. However, the key takeaway of the experiment is the additional and more fine-grained information provided by an image input over text alone. We argue this conclusion to be reasonable, given the fact that MultiFusion does indeed not replicate the input image but produces a variation with more aligned details. We modified the discussion of the experiment to reflect the strong supervision signal provided by the input image.
>
> Q6) Comparison to multi-lingual LMs
> This is an interesting comparison that we believe to be relevant for future work! We focused our empirical study on multilingual reasoning with MultiFusion in comparison to current state-of-the-art image generation approaches (as highlighted in section L209, Fig. 5a). At the same time, we believe our method to be robust enough to be reproduced with other multilingual LMs. However, we think that limitations may be encountered when using a bidirectional (e.g. mT5) instead of an autoregressive model, similar to the issue highlighted in response to 2Kwy [54].
>
> [4] Elad Ben Zaken, Yoav Goldberg, and Shauli Ravfogel. BitFit: Simple parameter-efficient fine-tuning for transformer-based masked language-models. In Proceedings of the Annual Meeting of the Association for Computational Linguistics, 2022.
> [16] Constantin Eichenberg, Sidney Black, Samuel Weinbach, Letitia Parcalabescu, Anette Frank. MAGMA - Multimodal Augmentation of Generative Models through Adapter-based Finetuning. EMNLP, 2022.
> [33] Niklas Muennighoff. Sgpt: Gpt sentence embeddings for semantic search.arXiv preprint arXiv:2202.08904, 2022.
> [54] Amirhossein Kazemnejad, Inkit Padhi, Karthikeyan Natesan Ramamurthy, Payel Das, and Siva Reddy. The Impact of Positional Encoding on Length Generalization in Transformers. arXiv preprint arXiv:2305.19466, 2023.

---

> > ### Comment · Reviewer_sjMA · 2023-08-21
> >
> > Thank the authors for the detailed information and updates included in rebuttal. Thanks for addressing all my comments and questions. I have read all the information and would like to keep the original score for recommendation.

---

### Official Review · Reviewer_8eHc · 2023-07-09

**Soundness:** 2 fair
**Presentation:** 2 fair
**Contribution:** 3 good
**Rating:** 6
**Confidence:** 4

**Summary:**

This paper presents an approach for creating a model that can take interleaved sequences of images and multi-lingual text as input, and generate novel images as output, by fusing together pre-trained models: (1) a ResNet image encoder from CLIP (2) an encoder-decoder text Transformer LM and (3) a Stable Diffusion (SD) image decoder. Most of the weights of the models are frozen, with some fine-tuning of adapter layers, the biases of the LM, and the cross-attention layers of the SD U-Net. Multimodal training is done on a combination of large scale image captioning, and VQA datasets, using the standard text-conditioned diffusion objective. The encoder-decoder text model was pre-trained on multi-lingual data, making the resulting image generation model also multi-lingual.

**Strengths:**

S1) The aim of this work, enabling an image generation model to take both images and text (in multiple languages) as input, is exciting and well-motivated by some of the qualitative examples in the paper: multi-modal inputs give complementary info, and multi-lingual text capabilities should broaden model accessibility.

S2) I found the experiments on compositional robustness (MCC-250), with the improved results from the combination of this text encoder and the image inputs, interesting and think it has the potential to be a timely addition to the ongoing conversation about the role of the pre-trained text-encoder in compositional robustness of image generation (but see suggestions on baselines below). Doing a human evaluation user study was also a real strength of these experiments.

S3) The qualitative results were compelling, particularly Figure 5 in the main text and Figure 4 in the appendix.

**Weaknesses:**

W1) The experimentation was a bit thin.
- Although there is definitely a shortage of current benchmarks for the new capabilities presented by this model, the contribution of the paper would be stronger if it were able to reappropriate existing benchmarks or create new ones to evaluate some of these capabilities (e.g. negative prompting with images, multimodal image composition).
- The method has a few steps (e.g. contrastive fine-tuning on a natural language inference dataset; training on a large number of multimodal datasets, both VQA and captions; and using attention manipulation), but I couldn't find any ablations on these components. This, in combination with the lack of details on the [apologies, the rest of this sentence was missing earlier] datasets, makes me worried about whether the overall approach will benefit future work.
- The quantitative results that are presented here would be more convincing with a few (hopefully) easy-to-run variants of the current settings (another classifier-free-guidance weight; ablating image inputs in the MCC-250 experiment); see questions below. The compositional robustness results are interesting, but giving an image as input is a pretty strong (and potentially unrealistic) source of supervision.

[update after response] : I still feel that point a) above, about capability evaluation, is a weakness, but the author response definitely helped address the other points. Thank you!

W2) The method relies on proprietary datasets and models for the language model (and possibly also for the image datasets, see questions below). I don't think this would be a crucial weakness except that almost no information is given about these datasets and models, even in the appendix. Given that the LM is frozen when doing the multimodal training, and that the capabilities of the fused system (with respect to multi-linguality, and the compositional robustness experiments) seem very likely to me to depend on the properties of this LM, more openness (ideally, using a publicly-released multimodal encoder-decoder transformer, like mT5-XXL, which also has 13B parameters) would really enhance the scientific value of this paper.
- The encoder is described as a "13B transformer encoder-decoder similar to GPT-3", but GPT-3 is a decoder-only model, trained with a language modeling objective.
- The LM dataset is described only as "400B tokens of English, German, French, Italian, and Spanish", and it's unclear whether the multimodal training data includes datasets other than the ones listed in lines 17-18 of the appendix.
- The German-English versions of SNLI and MNLI used for the semantic embedding objective also seem to be proprietary.

W3) The writing was somewhat unclear. In particular, a lot of details about the model (the pre-trained models used, the training data for the full approach) were unspecified in the main text, although outlined in the appendix (Section A). Some details about the experiments were also unclear, see questions.

[update after response]. The response effectively addressed both W2 and W3 -- thanks!

**Questions:**

Q1) It would be helpful to give any of the following results that are available:
- Scores for Table 1 with guidance scale 1.0, as SD seems to surpass MF as guidance scale decreases.
- Results for Table 2 that also remove the image input (i.e. use just the text), to see if there's a benefit in compositional robustness from using the pre-trained encoder-decoder model (this would be very cool if so!).
- Ablations of any components of the approach, e.g. the semantic embedding fine-tuning, the attention manipulation, or some of the datasets used in multimodal training (e.g., how much value do the VQA datasets add).

Q2) Could more information be given about the encoder-decoder LM, in particular what objective was it pre-trained with (e.g. a denoising objective? prefix-LM)? What is in the training data (e.g. web pages? books? is there paired data across languages, or all monolingual corpora)?

Q3) What data is used to train the multimodal adapters (to input images) and the SD U-net? It wasn't totally clear to me from the appendix. In particular, are other multimodal datasets used beyond the ones listed in lines 17-18 of the appendix? "such-as" and "like" make it seem like there could be others -- can you say anything about them, if so?

Q4) What image is being provided as input in the Table 1 results? Is it the ground-truth image? If so, could you give some intuition for this? The results are much better with multimodal and image, but ground-truth image would be really strong supervision (and probably also explain why multimodal is worse).

Not crucial for the author response, but it would also really improve the paper to clarify:
- How are the translations of SNLI/MNLI (line 24 of the appendix) generated?
- how is the contrastive training on the entailment dataset done?
- give details of the multimodal training (e.g. learning rates, dataset size, GPU hours) for both the adapters and the SD U-net.
- I didn't understand Fig 4a, as it seems that similarity needs a translation in two language pairs (e.g. en--de) but the categories here are single language.


**Limitations:**

I felt that the limitations section was pretty solid in qualitatively outlining weaknesses of the approach, although I'd appreciate experiments to quantitatively support the claim that attention manipulation can help prevent image context overriding text context.

---

> ### Author Rebuttal · Authors · 2023-08-10
>
> We thank the reviewer for their thorough review and insightful questions and will address them in the following. We strongly believe that the changes made during the review process have improved the quality of the paper.  [Rebuttal Pdf contains Tab. 5-7 as well as Fig. 11 and 12]
>
> ### Q1) Further Results
> 1. For guidance scale 1.0 StableDiffusion (SD) achieves a FID score of 26.09 and MultiFusion (MF)  achieves FID scores of 32.81 (Text), 24.22 (Multimodal) as well as 18.93 (Image). These results indeed show that SD does surpass MF for guidance scales <= 2. However, in line with previously reported results for Stable Diffusion, model performance significantly degrades for scales < 2. Further, CLIPScores for guidance scale 1.0 are 0.27 for SD and 0.25 for MF, which is in line with results for guidance scales 8.0 - 2.0. We will extend Tab. 1 with these results for the final version of the paper.
> 2. We appreciate the reviewer’s suggestion to further illustrate the benefit of multimodal over text-only input and extend Tab. 2 respectively. Due to the limited time and resources available during rebuttal, we conducted the user study on a smaller scale with 1 sample per prompt. For the final version of the paper, we will provide the complete study at full scale and extend Tab. 2.
>
> |Methods|Zero obj|One obj.|One obj. w/ correct color|Two obj.|Two obj. w/ correct colors|
> |--|--|--|--|--|-|
> | Stable Diffusion [%]| 0.49|99.50| 90.25| 45.63| 28.57
> | Composable Diffusion [%] | 2.93| 97.01| 88.83| 36.65| 25.12
> | MultiFusion [%]| 0.67| 99.32| 93.57| 62.37| 54.24|
> | MultiFusion (text) [%]| 0.8| 99.00| 80.00| 43.00| 24.00|
>
> One can observe that MultiFusion text-only is roughly on par with the other text-only models.
> Consequently, it is indeed the multimodal inputs and not the change in encoder architecture providing the performance increase.
>
> 3. Early experiments have shown higher rates of convergence (based on visual inspection of generated outputs) for experiments using semantic embeddings. Consequently, we do not report ablation results without semantic fine-tuning, as it is an essential condition and not an optional architecture choice for successfully fusing an image generation model. We adjusted section 3 of the paper accordingly to provide more clarity on this behavior.
> Similarly, Attention Manipulation is required to counteract the fact that images are encoded by an order of magnitude more tokens than short text prompts. We show representative examples of how attention manipulation strengthens the influence of the text prompt on the generated output in App D Fig. 9 as well as Fig. 11 of the rebuttal. Further, we compute additional FID scores (cf. Tab. 7) for the multimodal prompt ablating the attention manipulation weight on the text prompt, showcasing that the higher the weight the closer the FID score is to a text-only prompt. This quantitatively verifies that a higher attention manipulation weight on text prompts increases the influence on the generated image.
> Our multimodal dataset is based on the findings and extensive ablations of MAGA [16], thus we did not deem it crucial to perform our own dataset ablations.
>
> ### Q2) LM Details
>
> We thank the reviewer for pointing out the inconsistency in the abstract regarding the LMs architecture. As stated in the main body of the paper, MultiFusion uses an auto-regressive transformer, i.e. a decoder-only architecture similar to GPT-3. We adjusted the appendix accordingly. We supply further information on the training data sizes for all components in Tab. 5, along with details on data splits, languages, and modalities in Tab 6. We will add both tables to the final version of the paper.
>
> Only the semantic bias training uses paired multi-lingual data. All other training data is sourced from monolingual corpora.
> ### Q3) Multimodal training
>  The multimodal components (image prefix and adapters) are trained autoregressively on a combination of large-scale image-text datasets (c.f. Tab.5 and 6). Following MAGMA [16], the image tokens are prepended to the text tokens, and the language modeling loss for next token prediction is computed over the text only.
> The cross-attention parameters of SD are finetuned on the LAION aesthetics dataset following the standard diffusion objective. We added further clarification to the paper regarding these training details.
>
> ### Q4) Reference Image
> The reference image used for the experiment in Tab. 1 is indeed the ground truth image from COCO. We agree with the reviewer that this is strong supervision, most likely not available for a real-world use case. However, the key takeaway of the experiment is the additional and more fine-grained information provided by an image input over text alone. We argue this conclusion to be reasonable, given the fact that MultiFusion does indeed not replicate the input image but produces a variation with more aligned details. We modified the discussion of the experiment to reflect the strong supervision signal provided by the input image.
>
> ### Additional questions:
> The translations of SNLI and MNLI were automatically generated using the DeepL API, which we clarified in the appendix of the paper.
> Further training details are included in Tables 5 and 6
> The scores in Fig 4a) are indeed based on paired data. In this case, we report the average pair-wise similarity of the displayed language with all other languages. We adjusted the caption of the Figure accordingly to provide further clarification.
>
> [16] Constantin Eichenberg, Sidney Black, Samuel Weinbach, Letitia Parcalabescu, Anette Frank. MAGMA - Multimodal Augmentation of Generative Models through Adapter-based Finetuning. EMNLP, 2022.

---

> > ### Comment · Reviewer_8eHc · 2023-08-16
> >
> > Thanks to the authors for the detailed response, which addressed many of my concerns! I've raised my score from a 5 to a 6.

---

### Author Rebuttal · Authors · 2023-08-10

We thank all the reviewers for their detailed and helpful feedback.
We are encouraged that they found our solution for expressive image generation to be well-motivated (8eHc), novel (sjMA, g96C), and well-written (sjMA, 5v9N). Reviewers highlighted the efficient fusion of pre-existing models for simple, sample-efficient finetuning of a diffusion model (sjMA, 5v9N, 2Kwy), and improved capacity for flexible expression of complex concepts from combining complementary strengths in multimodal prompt interweaving (8eHc, sjMA, g96C, 2Kwy). We are pleased about the recognition of the importance and value of compositional robustness benchmarking (8eHc, g96c) as well as multilingual alignment evaluation (g96c).


Based on the reviewers’ suggestions, we provide further information on architectural choices, datasets as well as the overall training procedure. In the supplementary PDF, we supply further information on the training data sizes, parameter counts, and GPU hours for all components in Tab. 5, along with details on data splits, languages, and modalities in Tab 6. Further, we share additional qualitative (Fig. 11) and quantitative (Tab. 7) ablations on attention manipulation as well as qualitative examples of reversing the order of image inputs (Fig 12.). We consolidate common concerns and responses here and reply to the remaining comments individually in the hope to address them accordingly.


**@8eHc, sjMA, g96C, 2Kwy**
### Data:
We use proprietary datasets for both multimodal and LM training. However, we acknowledge that downstream capabilities are derived from these models and hope that the information in tables 5 and 6 can provide additional insight.
### Reiteration on architecture design:
Previous work has demonstrated that text encoders based on context-sensitive LMs improve the expressiveness of downstream image generation models [40,2]. Accordingly, we model the backbone of MultiFusion’s encoder as a 13B autoregressive transformer [10] trained on a multilingual corpus (cf Tab. 5 and 6). We justify our choice of an autoregressive decoder model over a bi-directional architecture by arguing that decoder models outperform bi-directional models on tasks such as manipulation with natural language (“Subject X with background Y) (cf qualitative results) or correct features attributions (“Red X, Blue Y”) (cf MCC-250 results). These capabilities have previously been attributed to the natural breaking of permutation equivariance [54], compared to bidirectional models relying entirely on positional embeddings.
Following the method proposed by MAGMA [16], we add an image prefix and dedicated adapters to enable multimodal capabilities. The adapters are added to each attention and feed forward layer of the transformer and are trained autoregressively on a combination of large scale image-text datasets (cf Tab. 5 and 6), while the parameters of the language model remain frozen. We argue that adapters are a suitable architectural choice for multimodal prompts with arbitrarily interleaved sequences of text and image tokens, as previous research has already performed extensive ablations on adapter architectures and demonstrated their improved understanding of multimodal inputs over other methods [16].
Our choice of semantic embeddings was guided by the intuition that a focus on the semantics of a text prompt would best capture the information relevant to image generation and thus simplify learning the mapping from embeddings to image outputs. We decided to obtain high-quality semantic embeddings through parameter-efficient bias [4] instead of full model finetuning, based on the work of [33]. The finetuning follows the supervised contrastive learning objective outlined in section 4.1.1 of [33].
In the final step we finetune the cross-attention parameters of SD on the LAION aesthetics dataset following the standard diffusion objective, which is in line with previous research [13].
We adjusted Fig. 1 and its caption, as well as section 3, to clarify details on the architecture and to better reflect the design choices.

**@8eHc, sjMA**
### Ablations:
***Attention Manipulation***: Attention Manipulation is required to counteract the fact that images are encoded by an order of magnitude more tokens than short text prompts. We show representative examples of how attention manipulation strengthens the influence of the text prompt on the generated output in App D Fig. 9 as well as Fig. 11 of the rebuttal. Further, we compute additional FID scores (cf. Tab. 7) for the multimodal prompt ablating the attention manipulation weight on the text prompt, showcasing that with increasing weight the FID scores approach that of text only prompting. The experiment empirically verifies that a higher attention manipulation weight on text prompts increases their influence on the generated image.

***Semantic Finetuning***: Early experiments have shown higher rates of convergence (based on visual inspection of generated outputs) for experiments using semantic embeddings. Consequently, we do not report ablation results without semantic fine-tuning, as it is an essential condition and not an optional architecture choice for successfully fusing an image generation model. We adjusted section 3 of the paper accordingly to provide more clarity on this behavior.

[2] Yogesh Balaji, et al. eDiff-I. arXiv:2211.01324
[4] Elad Ben Zaken, et al. BitFit. arXiv:2106.10199
[10] Tom Brown, et al. Language models are few-shot learners. arXiv:2005.14165
[16] Constantin Eichenberg, et al.. MAGMA. arXiv:2112.05253
[33] Niklas Muennighoff. Sgpt. arXiv:2202.08904
[40] Teven Le Scao, et al. BLOOM. arXiv:2211.05100
[54] Amirhossein Kazemnejad, et al The Impact of Positional Encoding on Length Generalization in Transformers.  arXiv:2305.19466, 2023.

---

### Decision · Program_Chairs · 2023-09-21

**Decision:**

Accept (poster)

**Comment:**

Paper received slightly divergent ratings of: 3 x Accept, 1 Weak Accept and 1 x Borderline Reject. Main concerns from reviewers focused on: (i) limited experimental validation, (ii) reliance on proprietary dataset, (iii) clarity of exposition, (iv) lack of novelty, and (v) lack of motivation for the problem. Provided rebuttal addressed many of these concerns and 4 out of 5 reviewers were left satisfied with responses. [g96C] remained somewhat unconvinced by the paper's contributions and robustness, however, did not strongly oppose acceptance of the paper. AC has looked at the reviews, rebuttal, discussion that followed and the paper itself. AC agrees with the more positive reviewers that paper is interesting, timely and would make a valuable contribution to the community. Therefore the recommendation is to Accept the paper. Authors are encouraged to incorporate their responses to the reviewers, along with additional experimental results provided in the rebuttal, into the main paper for the camera ready.